# Transferrin receptor 1 is a cellular receptor for human heme-albumin

Brell Jennifer[1,7], Verena Berg [1,7], Madhura Modak[1], Alexander Puck[1], Maria Seyerl-Jiresch[1], Sarojinidevi Künig[1], Gerhard J. Zlabinger [1], Peter Steinberger [1], Janet Chou[2], Raif S. Geha[2], Leopold Öhler[3], Akihiro Yachie[4], Hyeryun Choe [5], Markus Kraller[6], Hannes Stockinger[6] & Johannes Stöckl [1✉]

Iron is essential for living cells. Uptake of iron-loaded transferrin by the transferrin receptor 1 (CD71, TFR) is a major but not sufficient mechanism and an alternative iron-loaded ligand for CD71 has been assumed. Here, we demonstrate that CD71 utilizes heme-albumin as cargo to transport iron into human cells. Binding and endocytosis of heme-albumin via CD71 was sufficient to promote proliferation of various cell types in the absence of transferrin. Growth and differentiation of cells induced by heme-albumin was dependent on heme-oxygenase 1 (HO-1) function and was accompanied with an increase of the intracellular labile iron pool (LIP). Import of heme-albumin via CD71 was further found to contribute to the efficacy of albumin-based drugs such as the chemotherapeutic Abraxane. Thus, heme-albumin/CD71 interaction is a novel route to transport nutrients or drugs into cells and adds to the emerging function of CD71 as a scavenger receptor.

[1] Institute of Immunology, Center for Pathophysiology, Infectiology and Immunology, Medical University of Vienna, 1090 Vienna, Austria. [2] Division of Immunology, Boston Children´s Hospital, Boston, MA 02115, USA. [3] Department of Internal Medicine, St. Josef Hospital, 1130 Vienna, Austria. [4] Department of Pediatrics, School of Medicine, Institute of Medical, Pharmaceutical, and Health Sciences, Kanazawa University, Kanazawa, Japan. [5] Department of Immunology and Microbiology, The Scripps Research Institute, Florida, CA 92037, USA. [6] Institute of Hygiene and Applied Immunology, Center for Pathophysiology, Infectiology and Immunology, Medical University of Vienna, 1090 Vienna, Austria. [7] These authors contributed equally: Jennifer Brell, Verena Berg. ✉email: johannes.stoeckl@meduniwien.ac.at

ron is an essential and multifunctional element for almost all known forms of life[1–4]. Numerous enzymes involved in DNA replication, repair and translation rely on iron and therefore iron is particularly indispensable for proliferating cells. For example, DNA synthesis requires ribonucleotide reductase that has an obligatory requirement for iron[1]. Transferrin is an abundant plasma protein which binds iron and delivers it via binding and internalization through its receptor CD71 into cells. CD71 is absent or expressed at low levels on non-proliferating cells but is rapidly upregulated upon cellular activation and growth[1,5,6].

The transferrin/CD71 system is especially important for erythroblasts and lymphocytes, T and B cells. Accordingly, deficiencies in this major iron-uptake route lead to anemia and compromised immune responses[2,7]. Surprisingly, several studies have shown that defects of the receptor are more severe than defects or mutations affecting transferrin[7–9]. The reason for this discrepancy is unclear but it is intriguing that CD71 might be also used by other iron-carrying molecules.

Albumin is the most abundant protein in our blood. It has the unique ability to interact with numerous different substances such as lipids, drugs and also heme. Heme is iron-protoporphyrin IX. It is released from erythrocytes, mainly in the case of inflammatory or infectious diseases[10–14]. Since free heme is a potential toxic substance for cells, it is immediately scavenged in our body by plasma proteins including hemopexin, the predominant binding partner, and albumin. It has been reported that human serum albumin (HSA) has at least two binding sites for heme. One site of HSA binds heme with high affinity and this binding is not affected by fatty acids[15]. The other site of HSA binds heme with low affinity and is used by various hydrophobic molecules, including fatty acids[16,17]. The reported amounts of heme-albumin found regularly in our blood are in the range of $1 \times 10^{-6}$ M to $1.5 \times 10^{-6}$ M and this level is enhanced during infection or inflammation to up to $5 \times 10^{-5}$ M[18–20]. It is assumed that albumin is binding about 45% of heme in our blood and that up to 5% of albumin in humans is loaded with heme[21]. Given the abundance of HSA, a large amount of heme can be handled by HSA, which might serve as an alternative, or perhaps additional mechanism to provide cells with iron.

## Results

**HSA-heme is able to promote cell proliferation.** In order to test whether HSA-loaded with hemin (HSA-heme) can be used by proliferating cells as an iron-source, we cultured Jurkat T cells with recombinant HSA or recombinant HSA-heme in serum-free and protein-free medium. Results presented in Fig. 1a and Supplementary Figs. 1 and 2 demonstrate that supplementing medium with recombinant HSA-heme but not recombinant HSA promotes proliferation of the cells. In contrast, HSA-loaded with biliverdin (Fig. 1a) or with protoporphyrin IX (Supplementary Fig. 3) is not sufficient for cell growth. The same proliferative response was observed with primary human T cells upon stimulation via CD3/CD28 or permanent cell lines such as erythroblast TF-1 cells, monocytic THP-1 cells and epithelial Hela cells (Fig. 1b, Supplementary Fig. 4). Addition of hemin alone in protein-free medium was not sufficient to induce proliferation in cells (Supplementary Fig. 5). HSA from plasma but not bovine serum albumin (BSA) was also capable of promoting cell proliferation, however, higher concentrations of the protein were needed compared to heme-loaded HSA (Fig. 1c). This effect was observed with plasma-derived HSA from different sources, which carried heme but did not contain detectable amounts of transferrin (Fig. 1d, Supplementary Figs. 6, 7). Treatment of HSA with charcoal largely removed heme from the protein and HSA lost its ability to promote cell proliferation (Fig. 1d, Supplementary

Fig. 6). Labeling of plasma-derived HSA with heme strongly upregulated the proliferation-promoting capacity, whereas treatment of HSA with different lipids did not enhance its proliferative quality (Supplementary Fig. 8). HSA-heme is as potent as iron-loaded transferrin in promoting cell proliferation and there is no synergistic effect when both factors were applied in combination, suggesting that HSA-heme and transferrin may use the same transport route (Fig. 1e).

**CD71 is the receptor for HSA-heme.** To gain insight into a potential role of CD71 in the binding and uptake of HSA-heme, we used murine Bw-cells expressing human CD71 molecules and analyzed the impact of HSA-heme on the proliferation of such cells. Results presented in Fig. 2a demonstrate that recombinant HSA-heme but not HSA alone facilitated the proliferation of murine BW-cells transfected with human CD71 but not of control cells. Accordingly, murine splenocytes expressing mutated, human CD71 receptors, which cannot internalize their cargo[2,7], failed to proliferate in response to HSA-heme (Fig. 2b). Supplementation of culture medium with exogenous iron in form of ferric ammonium citrate (FAC) abrogated the need for HSA-heme in our protein-free culture condition (Fig. 2c). As demonstrated in Fig. 2a, c, the proliferative responses of CD71-transfected Bw cells, as well as of Jurkat T cells could be inhibited with mAbs against CD71. MAbs 5-528, 15-221 and VIP1 recognize human CD71 expressed on murine Bw cells (Supplementary Fig. 9). Blocking of CD71 with mAb VIP1 stopped proliferation induced by HSA-heme or of transferrin at the S/G2 phase of the cell cycle (Supplementary Fig. 10).

Using FITC-labeled proteins we could show that binding of HSA-heme to Jurkat T cells was blocked with CD71 mAb VIP1 and mAb 5-528, which recognize an overlapping epitope, but not with CD71 mAb 15-221 directed against a different epitope (Fig. 3a, b and Supplementary Fig. 11). In addition, Jurkat T cells lacking CD71 on their cell surface could not bind HSA-heme (Fig. 3c). CD71 is not only a receptor for transferrin but also for Machupo virus, which binds via GP1[22,23]. In order to investigate if HSA-heme uses similar or different binding sites on CD71 compared to the other ligands of the receptor, we performed binding studies and observed that transferrin reduces the binding of HSA-heme to Jurkat T cells whereas MACV-GP1Δ-Ig (Machupo virus glycoprotein) did not affect binding of HSA-heme and was inhibited by mAb 15-221 but not VIP1 (Fig. 3d, Supplementary Fig. 12). Thus, HSA-heme and transferrin have overlapping binding sites on CD71 which are different from the epitope used by MACV-GP1Δ-Ig. Results show in Fig. 3e, f demonstrate the internalization of HSA-heme in Jurkat T cells and a comparison with transferrin (Fig. 3e). Furthermore, proliferation induced by HSA-heme could be prevented with inhibitors of clathrin-mediated endocytosis (MiTMAB, Dynasore, Pitstop 2) (Fig. 3g).

**Heme oxygenase 1 is required for the utilization of HSA-heme.** Cells use heme-oxygenase 1 to liberate iron from heme[4,24]. In order to elucidate whether HO-1 is required that cells can use HSA-heme as an iron-source, we used lymphoblastoid YK01 cells which are HO-1 deficient[25]. YK01 cells express high levels of CD71 receptors like other lymphoblastoid B cell lines and proliferation in the presence of iron-loaded transferrin is inhibited by a CD71 mAb (Supplementary Figs. 13, 14). However, supplementing the protein-free medium with HSA-heme is not able to promote growth of YK01 cells in contrast to lymphoblastoid cells OTHAKA with express intact HO-1 (Fig. 4a). Cell proliferation in the presence of HSA-heme induces *HO-1* expression (Fig. 4b). The central role of HO-1 and the release of iron from HSA-heme

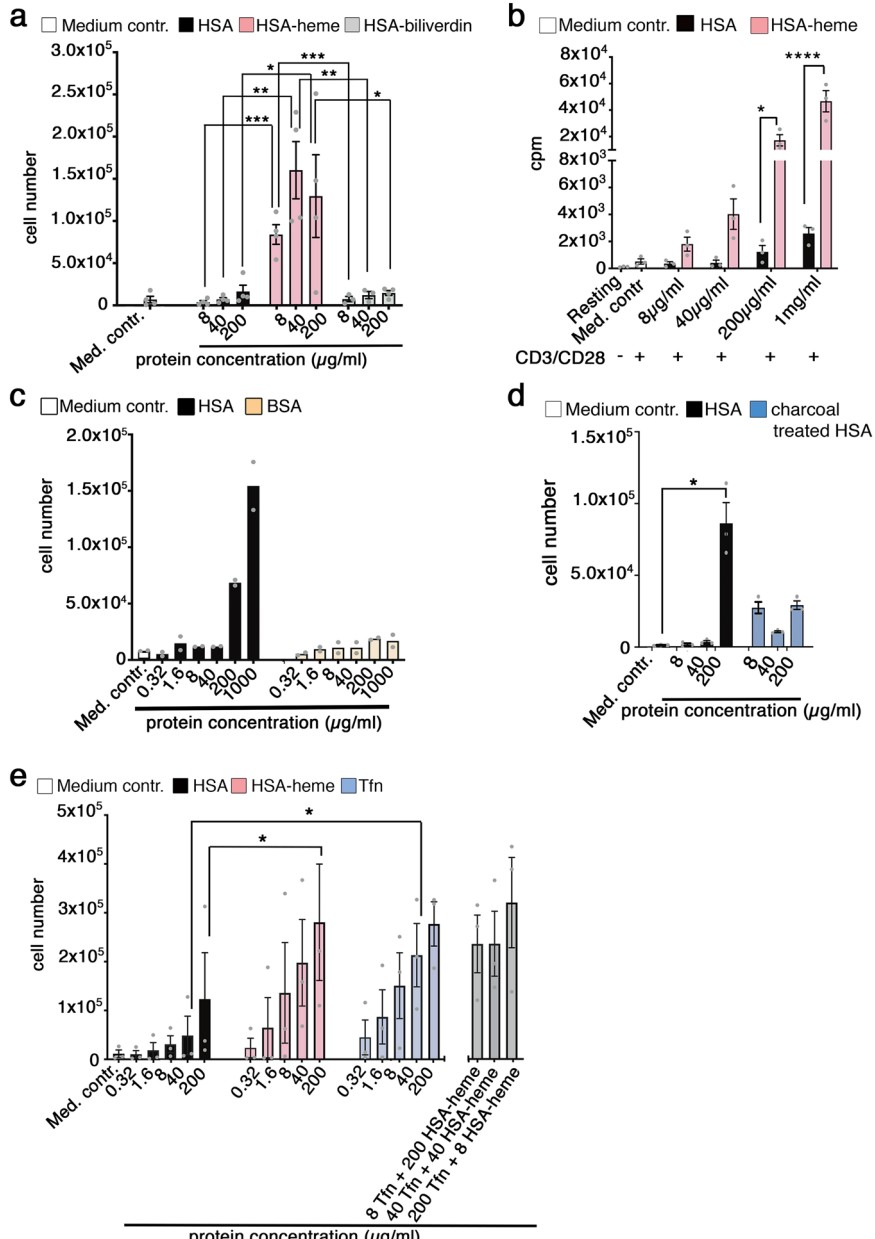

**Fig. 1 HSA-heme is sufficient to promote proliferation under serum-free and protein-free culture conditions. a** Effect of HSA-heme on the proliferation of Jurkat T cells. Cells were cultured in serum-free and protein-free medium supplemented with different proteins. Serum-free and protein-free RPMI medium was used as medium control (negative control) (number of experiments = 4). **b** Effect of HSA-heme on the proliferation of primary human peripheral blood T cells. T cells were activated via plate-bound mAbs CD3/CD28 and cultivated in medium supplemented with HSA or HSA-heme. Proliferation was measured after 4 days by analyzing [methyl-$^3$H]thymidine incorporation ($n = 3$). **c** As in (**a**) the proliferation of Jurkat T cells was analyzed in presence of HSA or BSA ($n = 2$). **d** Jurkat T cell proliferation was analyzed in presence of HSA or charcoal treated HSA ($n = 3$). **e** Proliferation of Jurkat T cells in presence of HSA, HSA-heme and transferrin ($n = 3$). In addition, a counter-titration with transferrin and HSA-heme in combination was performed ($n = 5$). **a–e** Results are displayed standard error of the mean (SEM). P values were calculated by using one-way ANOVA, followed by Tukey's multiple comparison test. *$P < 0.05$, **$P > 0.01$, and ***$P < 0.001$, ****$P < 0.0001$.

was further examined by the use of an inhibitor. Results presented in Fig. 4c demonstrate that proliferation of Jurkat T cells in the presence of HSA-heme but not fetal calf serum (FCS) is inhibited by Tin Protoporphyrin, an inhibitor of HO-1.

In order to analyze whether the levels of the intracellular pool of LIP is altered in Jurkat T cells in the presence of HSA-heme, we used a calcein-based method[26–28]. We observed that HSA-heme enhanced the amounts of LIP like the treatment of Jurkat T cells with FAC (Fig. 5a). Moreover, HSA-heme-mediated cell proliferation can be blocked with iron-chelator 311, which is cell

membrane permeable, but not with ethylenediamine tetraacetic acid (EDTA), which primarily acts extracellularly (Fig. 5b)[29–32]. The increase of intracellular iron affected the expression of the genes, which are regulated by the level of iron. Results presented Fig. 5c demonstrate that *TFR1 (CD71)* and *iron regulatory protein 1 (IRP1)* are downregulated in the presence of HSA-heme in Jurkat T cells, whereas *ferritin* is not significantly regulated, like we have observed in the case of adding iron in form of FAC. At the protein level, HSA-heme induced a downregulation of TFR1 (CD71) expression but an upregulation of ferritin expression in

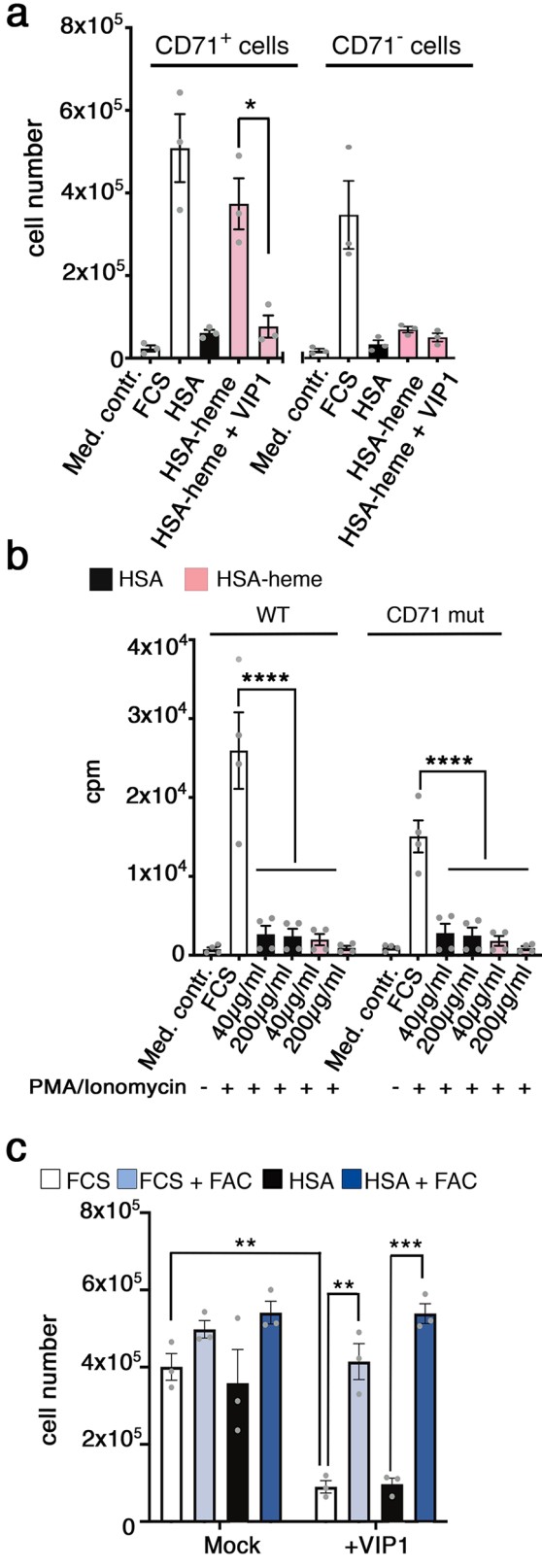

**Fig. 2 CD71 is the receptor for HSA-heme. a** Murine Bw cells lacking or expressing human CD71 receptors were cultured with 200 µg/ml HSA or HSA-heme in presence or absence of CD71 mAb VIP1 (10 µg/ml; $n = 3$). RPMI medium was used as medium control. **b** Proliferation of mouse spleenocytes, expressing mutated human CD71 (CD71mut), which cannot internalize their cargo. Spleenocytes were cultivated in RPMI medium supplemented with 10% FCS, HSA, or HSA-heme. In addition, cells were stimulated with PMA and Ionomycin and analyzed on day 5 via [methyl-$^3$H] thymidine incorporation ($n = 4$). **c** Effect of CD71 mAb VIP1 on the proliferation of Jurkat T cells. Cells were cultivated with 10% FCS, HSA, 10% FCS with FAC or HSA with FAC treated with or without VIP1 ($n = 3$). **a**, **c** Means ± SEM are shown. $P$ values: *$P < 0.05$, **$P > 0.01$, and ***$P < 0.001$ (two-tailed unpaired $t$-test). **b** Means ± SEM are shown. ****$P < 0.0001$ (one-way ANOVA followed by Tukey's multiple comparison test).

via CD71 is not only an inert nutrient and iron-source for cells but may influence cellular functions via down-stream degradation products of heme. Using a Jurkat reporter cell line, termed triple parameter reporter (TPR), we next analyzed whether HSA-heme influences three prominent signaling pathways (NF-κB, AP1, NFAT)[35]. Results presented in Fig. 6a demonstrate that Jurkat cells activated via CD3/CD28 in the presence or absence of plasma-derived HSA or HSA-heme turn-on NF-kB, AP1 and NFAT like the cells do in the presence of transferrin. HO-1 generated degradation products of heme are also known to affect the differentiation and function of the myeloid cell linage[36,37]. Myeloid cells, such as dendritic cells, upregulate CD71 expression upon differentiation and activation[5,6]. We observed that HSA-heme regulates the functional differentiation of monocyte-derived DCs. DCs generated in the presence of HSA-heme expressed lower levels of the typical marker CD1a compared to DCs generated in the presence of transferrin or FCS (Fig. 7a). The induction of production of several cytokines analyzed in this study (IL-12, IL-10, IL-1, and TNF) induced by LPS stimulation was reduced in HSA-heme cultured DCs, in particular IL-10 and IL-12p40 (Fig. 7b).

**The toxic effect of Abraxane is part mediated via CD71.** Albumin is a multifunctional protein and a potential carrier and scavenger of many different compounds such as lipids and drugs[38–40]. Human plasma-derived albumin is therefore applied as carrier and stabilizer in many formulations of medicines. Abraxane is an HSA-based drug that consists of paclitaxel, a chemotherapeutic agent, which is packed into HSA nano-particles[40,41]. Abraxane contains heme-moieties (Fig. 8a). Based on our observations we asked whether the CD71 route, which is also used by large molecular complexes such as viruses, may be involved in the uptake of Abraxane. The drug is toxic for Jurkat T cells down to 5 ng/ml (Fig. 8b). The toxic effect of Abraxane is caused by preventing microtubule depolymerization[42]. Reactivity of anti-beta-tubulin antibody is enhanced in Abraxane-treated cells and this effect is inhibited by CD71 mAb 5-528[43] (Fig. 8c). Moreover, blocking of CD71 with mAb reduces the number of dying cells by approximately 25% (Fig. 8d). Collectively, these results reveal a novel role of HSA-heme and CD71 in delivering iron to cells and suggest an impact of the scavenger function of CD71 in the mechanism of HSA-based drug applications.

Jurkat T cells (Fig. 5d). Thus, HSA-heme can provide cells with iron from heme catabolism involving HO-1.

**Impact of HSA-heme on cellular signaling and differentiation.** HO-1 degrades heme into biliverdin which is then further degraded into bilirubin and other down-stream products including CO[24,33,34]. Hence, we asked if HSA-heme transported

## Discussion
Albumin is somehow a forgotten stepchild, although it is the most abundant intravascular protein and a substantial extravascular protein. HSA is responsible for 80% of the colloidal osmotic pressure of blood. It is surprising that in spite of this impressive

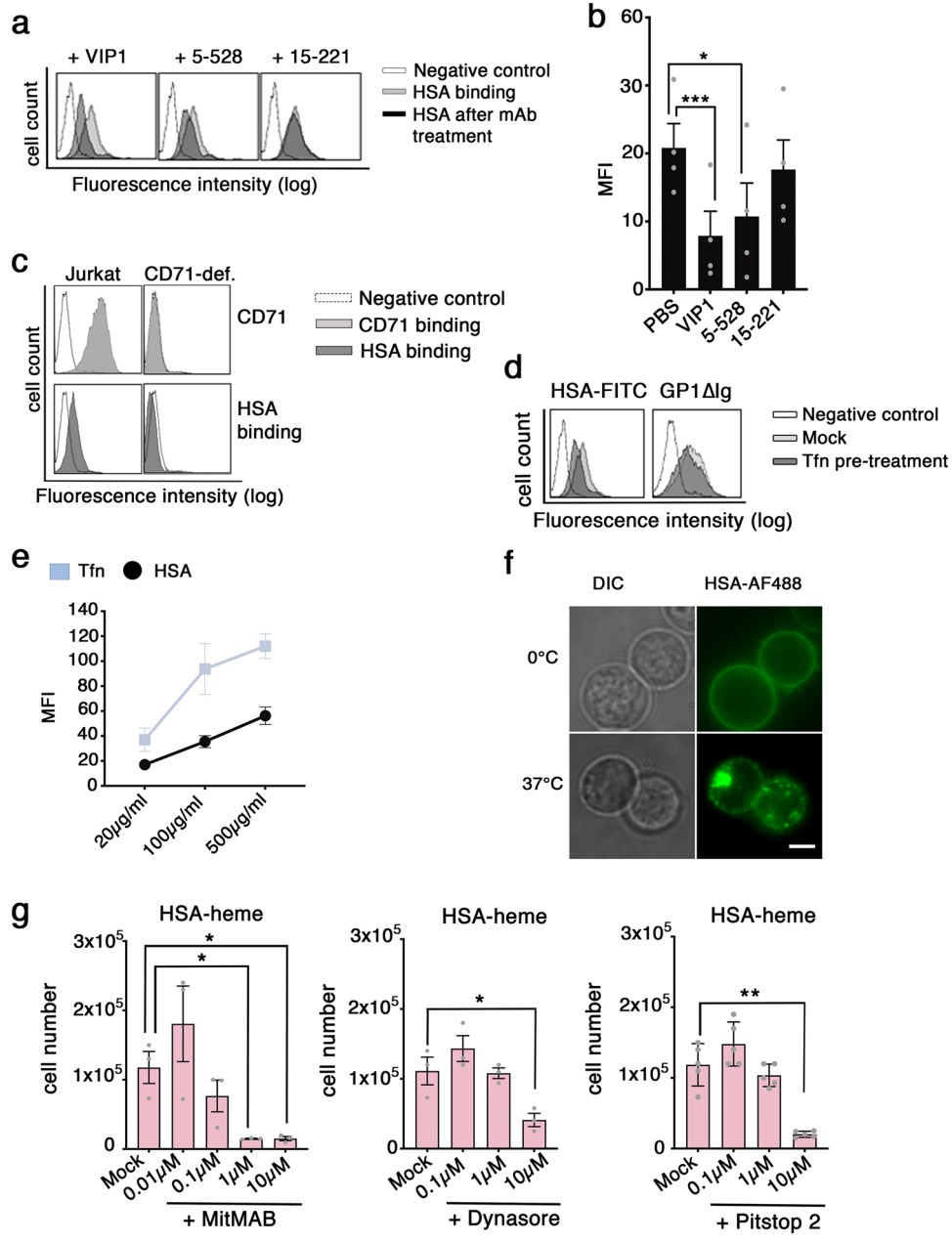

**Fig. 3 HSA-heme uptake is mediated by endocytosis. a** Effect of CD71 mAbs on HSA binding to CD71. Binding of FITC-labeled HSA (100 μg/ml) to CD71 was analyzed by flow cytometry. Jurkat T cells were pre-treated with or without CD71 mAb VIP1, 5-528, or 15-221. **b** Bar graph shows the mean fluorescence intensity (MFI) of HSA binding to CD71. Data shown are representative of four independent experiments. **c** Analysis of the binding of HSA-FITC to CD71 in Jurkat T cells and Jurkat T cells, which lack CD71 surface expression due to mAb-induced down-modulation. Fluorescence was measured by flow cytometry. Data presented is representative of three independent experiments. **d** FACS analysis of the binding and blocking of HSA and GP1ΔIg protein to CD71. Jurkat T cells were pre-treated with transferrin and incubated with FITC-labeled HSA (200 μg/ml) or GP1ΔIg (6 μg/ml) protein, followed by a second step staining with a secondary antibody. Data shown is representative for four independent experiments. **e** FACS analysis of the internalization of HSA-FITC and Tfn-FITC at 37 °C ($n = 4$). **f** HSA internalization was visualized by Epifluorescence microscopy on live cells. The uptake (37 °C) and binding (0 °C) of AF488-labeled HSA was assessed. Differential interference contrast (DIC) and fluorescence micrographs are shown. Results represent one of three independent experiments. Scale bar: 5 μm. **g** Jurkat T cells were cultivated in HSA-heme at a concentration of 200 μg/ml. In comparison to the control (Mock), cells were additionally treated with inhibitors MiTMAB ($n = 3$), Dynasore ($n = 3$), or Pitstop 2 ($n = 5$) at different concentrations. **b, g** Data show mean ± SEM. $P$ values were calculated by using one-way ANOVA, followed by Tukey's multiple comparison test. *$P < 0.05$, **$P > 0.01$, and ***$P < 0.001$.

plethora of functions, people suffering from hypo-albuminemia or from mutated HSA, have seemingly no serious health problem. These observations also demonstrate that HSA is obviously not of a single importance for our body and may rather serve as a molecular back-up system for several molecular functions.

Accordingly, although hemopexin is considered as the primary scavenger of free heme molecules, the high amounts of HSA molecules guarantee an efficient buffer system of the toxic effects of free heme on cells and HSA binds about 30% more free heme than hemopexin[11,21].

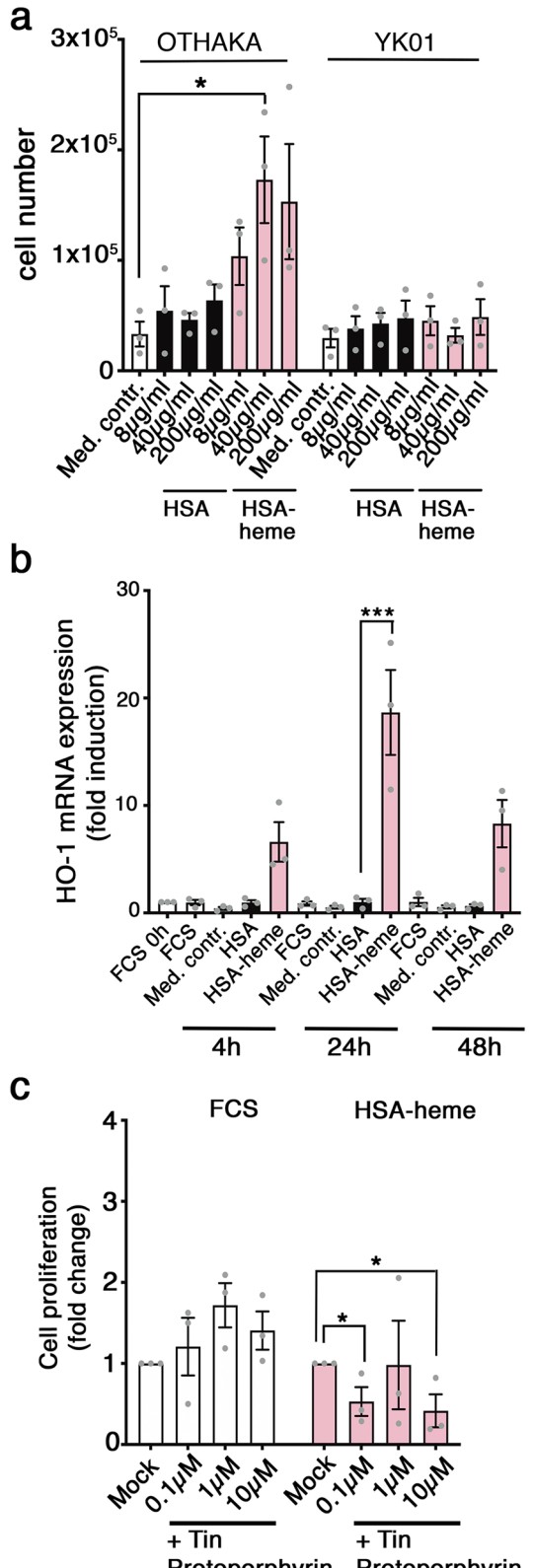

**Fig. 4 Utilization of HSA-heme by proliferating cells requires heme oxygenase 1 (HO-1). a** Proliferation of Epstein-Barr-Virus (EBV)-immortalized B cells, a wildtype (OTHAKA) and a cell line with a defect heme oxygenase 1 enzyme (YK01) in presence of HSA or HSA-heme ($n = 3$). RPMI medium without protein or serum was used as medium control. **b** HO-1 gene expression in Jurkat T cells. Levels of the indicated HO-1 mRNAs in the presence of HSA or HSA-heme (200 µg/ml) were quantified via qPCR after 4 h, 24 h, and 48 h. For each time point three independent experiments are shown. **c** Jurkat T cells were incubated in serum-free and protein-free medium supplemented with 10% FCS (Mock) or HSA-heme at a concentration of 200 µg/ml. Cells were treated with Tin Protoporphyrin (HO-1 inhibitor) in three different concentrations ($n = 3$). **a–c** Means ± SEM are shown. *$P < 0.05$, ***$P < 0.001$, (one-way ANOVA followed by Tukey's multiple comparison test).

uptake of HSA-heme can be utilized by different cell types to gain access to an alternative source of iron sufficient to promote their proliferation. We have shown this effect for primary human T cells, as well as for a panel of immortalized cell lines. Yet, it is intriguing to speculate that different cell types may differ in their capacity to catabolize and utilize HSA-heme as nutrient. Intact HO-1 function is important for cells to utilize iron after uptake of HSA-heme (Fig. 4a–c). Yet, degradation of heme by HO-1 is also an important cellular differentiation signal. We have observed in this study that HSA-heme is able to modulate the differentiation of monocyte-derived DC and to inhibit their cytokine production repertoire (Fig. 7b). So, it needs to be determined in future studies whether provision of iron via HSA-heme/CD71 mediated endocytosis in a general phenomenon in our cells and how this pathway may influence the functional behavior of cells.

CD71 is considered today as a promiscuous cell entry carrier. Its primary function is the import of iron bound to transferrin but several other ligands including ferritin, arenaviruses or malaria parasite use CD71 to enter cells. Our study demonstrates that HSA-heme is another intrinsic complex that utilizes this special cell entrance. The dissociation constant ($K_d$) value of HSA-heme binding is $7.52 \times 10^{-7}$ M, which is lower range of what has been reported for transferrin. The $K_d$ for bound diferric transferrin ranges from $10^{-7}$ M to $10^{-9}$ M at physiologic pH, depending on the species and tissue. Based on our binding studies, we conclude that the binding site of HSA-heme is in proximity to the transferrin binding site, which is located on the lateral part of the receptor ectodomain. This binding site is distinct from the ferritin/pathogen contact region of CD71, which is located on the apical region of CD71 ectodomain. Interestingly, the HSA-heme/CD71 route seems to be specific for human albumin and human CD71. Our results of this study demonstrate that plasma-derived BSA, which is also loaded with heme although not so frequent as human albumin[44,45], did not promote cell proliferation in our test system. Moreover, the data of our study obtained with murine cells revealed that HSA-heme can obviously not be used as an iron source (Fig. 2a, b). Thus, it seems as if the albumin/CD71 interaction could be species specific.

Iron is one of the central chemicals in our body and we demonstrate here a novel route of iron-uptake via CD71. However, we have to envisage that CD71 molecules exert also other, non-canonical functions. Mouse genetic studies have implicated iron-independent roles of CD71 in the early development of the nervous system[46], maintenance of intestinal cell homeostasis[47] and development of lymphocytes[48]. In addition, CD71 has signaling functions, including the activation of NF-κB and regulation of JNK[35]. It will be worth testing whether HSA-heme, an alternative ligand of CD71, may be involved in these non-canonical roles of CD71. Interestingly, the function of CD71 can

Our body and cells have to remove HSA-heme from the system. Several receptors for HSA have been reported for various cell types[13]. Here we demonstrate that CD71 is an important and specific cellular receptor for HSA-heme. The CD71/transferrin system is famous for iron-transport into proliferating cells and is a prototypic endocytic receptor, which is recycling to the cell surface. The results of our study demonstrate that binding and

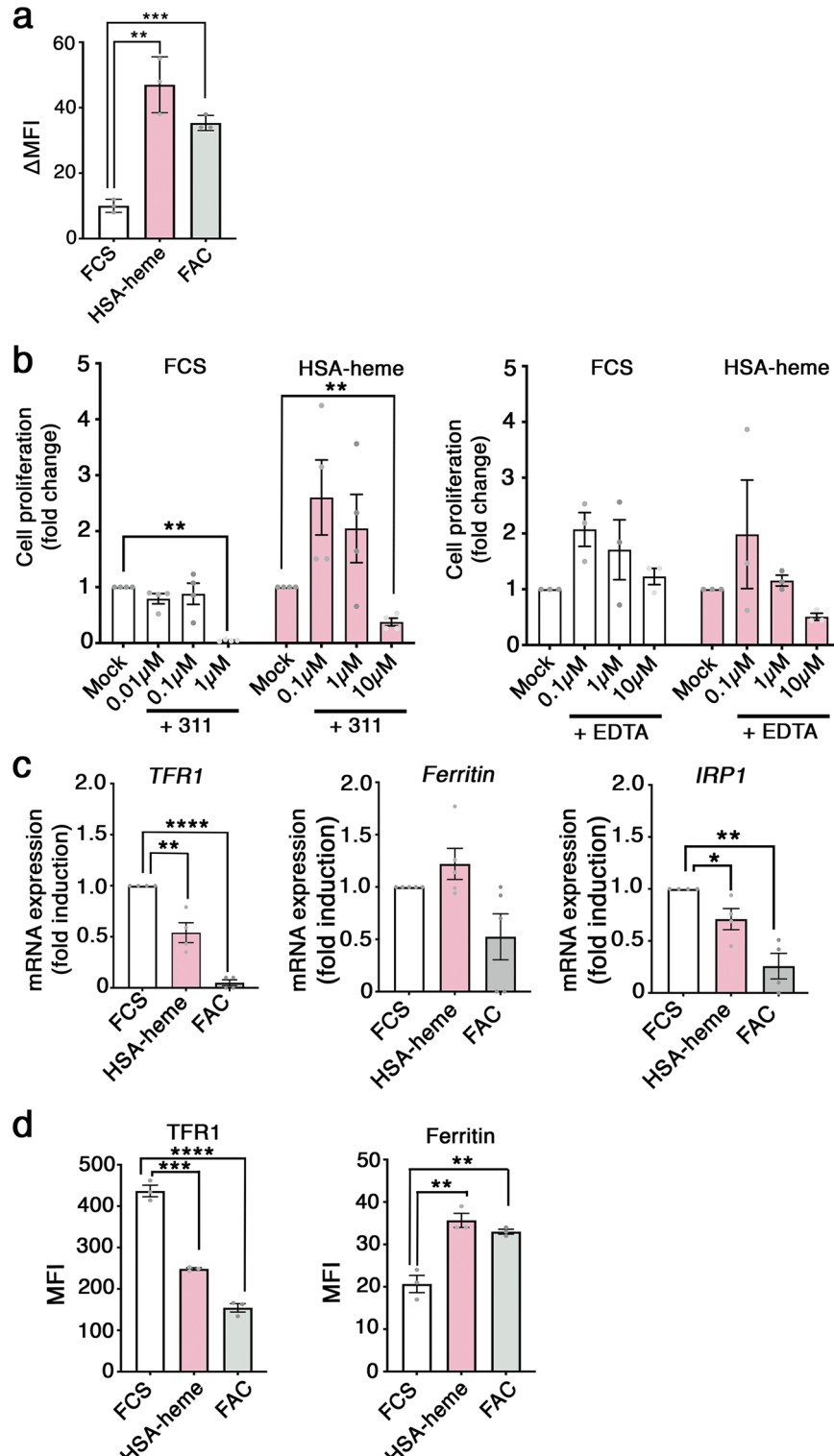

**Fig. 5 Iron from HSA-heme is used for cell proliferation. a** Impact of HSA-heme on intracellular levels of the labile iron pool (LIP). Jurkat T cells were incubated for 2 h with HSA-heme or FAC. Cells were loaded with Calcein-AM, washed and incubated with a combination of iron chelators: 311 ($Fe^{3+}$ chelator) and BIP ($Fe^{2+}$ chelator). Data show mean fluorescence between chelator-treated and untreated cells (Δ MFI). **b** Jurkat T cells were incubated in medium supplemented with 10% FCS (Mock) or HSA-heme at a concentration of 200 μg/ml. In addition, cells were treated with iron chelator 311 ($n = 4$) or EDTA ($n = 3$) at different concentrations. **c** TFR1, IRP1 and ferritin mRNA expression under different conditions. Jurkat T cells were incubated with 10% FCS, HSA-heme (200 μg/ml) or 10% FCS with FAC (25 μg/ml) for 6 h. Expression of mRNAs were quantified via qPCR and mRNAs were normalized to β2 m mRNA. Results are from three (TFR1, IRP1) or four (ferritin) independent experiments. **d** Expression of human TFR (CD71) and ferritin at the protein level. As in (**c**) Jurkat T cells were incubated with HSA-heme or FAC and protein expression was detected after 24 h (TFR) or 48 h (ferritin) by intracellular staining with CD71 mAb and anti-ferritin antibody and analyzed by flow cytometry. Data show mean values of MFI**. a–d** Means ± SEM are shown. **P > 0.01, ***P < 0.001, ****P < 0.0001 (two-tailed unpaired t test).

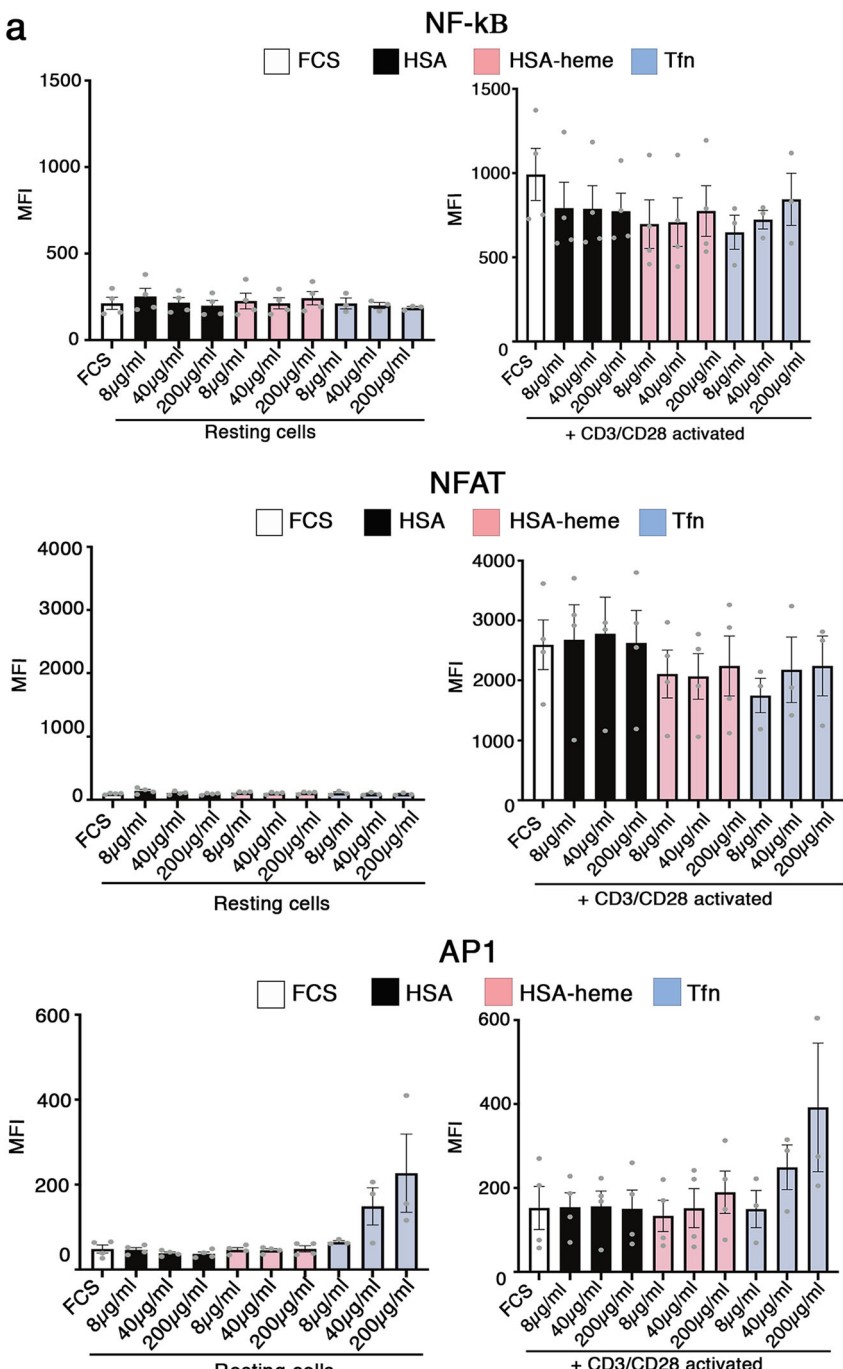

**Fig. 6 Impact of HSA-heme on cell signaling. a** A multi-channel reporter cell line from Jurkat T cells expressing reporter genes under control of NF-κB, NFAT, and AP-1 promoter elements were used to study the impact of HSA-heme on prominent signaling pathways. Bar graph shows the expression of the transcription factors NF-κB, NFAT, and AP1 in presence of 10% FSC, HSA, HSA-heme or transferrin analyzed by flow cytometry. Bar graph show resting cells and cells which were stimulated with a combination of CD3 and CD28 antibodies (5 µg/ml each). Graph show means ± SEM of MFI ($n = 4$). Only significant differences are indicated (one-way ANOVA followed by Tukey's multiple comparison test).

be modulated by stearoylation, which regulates the morphology of mitochondria[49] or by c-Abl kinase, which controls the endocytic fate of CD71[50]. This finding indicates that the interaction partners of CD71 may regulate its functional repertoire. HSA-heme has peroxidase-activity[51]. Peroxidase-activity is a well-known anti-microbial tool of our innate immune system[52–54]. Binding of HSA-heme to CD71 may thus decorate this important portal of viruses and malaria parasites with peroxidase activity. It is intriguing to speculate that since some of these viruses, such as

Machupo virus, cause hemorrhagic fever, the elevated levels of HSA-heme in such a situation may contribute to protect the CD71 entry site.

Albumin is an important drug carrier is gaining increasing importance in the target delivery of cancer therapy, particularly as a result of the market approval of the paclitaxel-loaded albumin nanoparticle, Abraxane[40,41]. CD71 is an entry gate for large molecular particles such as viruses[55] The results of this study that approx. 25% of the toxic efficacy of Abraxane is mediated via the

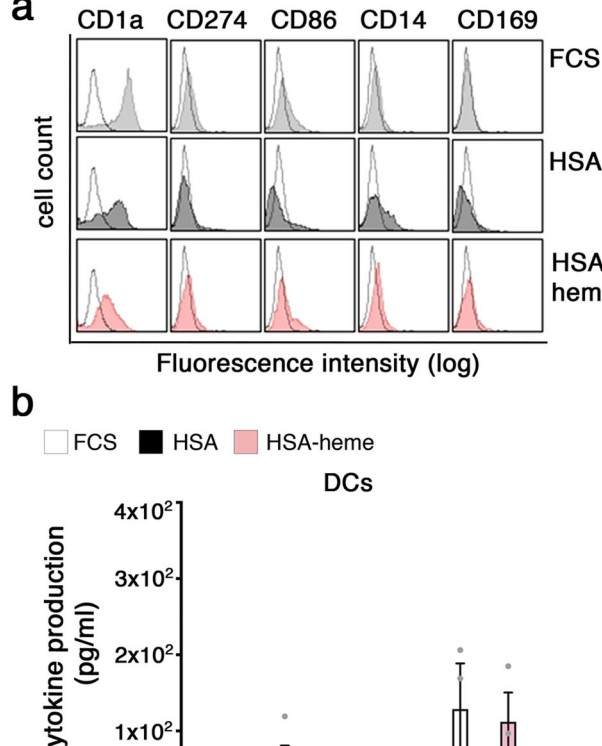

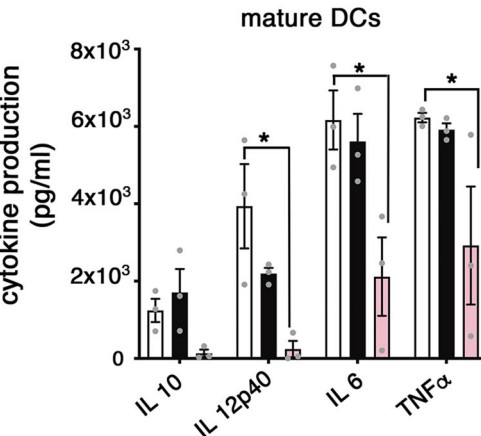

**Fig. 7 Impact of HSA-heme on the differentiation of monocytes into DCs. a** Phenotype of DCs generated in the presence of HSA-heme. Monocytes were differentiated into DCs in presence of 10% FCS, 1% HSA or 1% HSA-heme and analyzed after 7 days for expression of typical cell surface markers (filled histograms) compared to the isotype control (dotted histograms) via flow cytometry. Data shown are representative of four independent experiments with four different donors. **b** HSA-heme modulates cytokine production of DC upon stimulation with LPS (mature DCs). Supernatants were collected from DCs and mature DCs and analyzed on day 7 (n = 3). Means ± SEM are shown. *$P < 0.05$ (one-way ANOVA followed by Tukey's multiple comparison test).

herein described uptake mechanism of HSA-heme via CD71. Accordingly, we could demonstrate that CD71 mAb 5-528 reduces the increased beta-tubulin staining in cells upon Abraxane-treatment. The binding sites of mAb 5-528 and VIP1 are overlapping (Supplementary Figs. 9 and 10). Both mAbs inhibit binding of HSA-heme to CD71, where 5-528 is less efficient (Fig. 3a, b). However, mAb VIP1 inhibits cell proliferation (Fig. 2), whereas mAb 5-528 does not. Thus, the inhibitory mAb VIP1 is not suitable to revert the cell toxic effect of Abraxane, whereas 5-528 is able to partly revert the Abraxane killing because the mAb is not affecting the proliferation and viability of the cells per se.

Taken together, the HSA-heme/CD71 route described in this study offers new insights of iron-uptake of cells, the design of future HSA based drugs and the usage of HSA as plasma-compensation.

## Methods

**Cell lines and culture**. All the cell lines used in this study were tested for mycoplasma contaminations using a method described before[56]. Jurkat T cells (TIB-152), TF-1 cells (CRL-2003), THP1 cells (TIB-202), HeLa cells (CCL-2), Bw5147 cells (TIB-47) were obtained from ATCC (American Type Culture Collection, Virginia, US). A multi-channel reporter cell line from Jurkat T cells expressing reporter genes under control of NF-κB, NFAT and AP-1 promoter elements and murine Bw cells transfected with human CD71 were generated at our institute as described[37,57]. The EBV-transformed lymphoblastoid B cell lines YK01 (HO-1 deficient) and OTHAKA were generated as described before[25,58]. Jurkat T cells which do not express CD71 on the cell surface (CD71 deficient Jurkat T cells) were generated due to down-modulation of cell surface molecules by mAb VIP1 treatment and subsequent culturing in the presence of exogenous iron in form of FAC. All cell lines, were cultured in RPMI 1640 medium (Invitrogen, Paisley, United Kingdom), supplemented with 2 mM L-glutamine, 100 U/ml penicillin, 100 µg/ml streptomycin and 10% FCS purchased from HyClone Laboratories Inc. (Utah, US), in a humidified atmosphere (5% $CO_2$) at 37 °C. In case of TF-1 cells 2 ng/ml GM-CSF was additionally added to RPMI 1640 medium. All of the above-mentioned cell lines are immortalized or tumor-derived cell lines.

**Antibodies and reagents**. The following murine monoclonal antibodies (mAb) were raised in our laboratory: negative control mAb VIAP (against calf intestine alkaline phosphatase), 5-528 (CD71), VIP1 (CD71)[59,60], 15-221 (CD71), 13-344 (transferrin), mAb VIT6b (CD1a-PE), 5-272 (CD274-PE), MEM 18 (CD14-PE), and 7-239/44/0 (CD169-PE). APC-conjugated donkey anti-human IgG, goat anti-human IgG and anti-mouse conjugated alkaline phosphatase antibodies were purchased from Jackson-Immunoresearch Laboratories Inc. (Newmarket, UK). mAb OKT3 (CD3) was obtained from Janssen-Cilag (Vienna, AT). Human granulocyte–macrophage colony-stimulating factor (GM-CSF) and IL-4 were kindly provided by Novo Nordisk A/S (Bagsværd, DNK). MAb B7-2 (CD86-PE), mAb 10F3 (CD28), Oregon Green 488-conjugated goat anti-mouse IgG antibody and UltraPure EDTA was obtained from Invitrogen, UK. Native HSA-FITC, MiTMAB and Deferiprone were purchased from Abcam PLC (Cambridge, UK). Anti-human beta-tubulin antibody (9F3-AF488) was acquired from Cell Signaling Technology Inc. (Frankfurt, DEU). Anti-human ferritin antibody (heavy chain, ferritin-AF647) was purchased from Santa Cruz (Dallas, US) and anti-human TFR antibody (TFR-AF647) from BD (Vienna, Austria). The following products were purchased from Sigma-Aldrich (St. Louis, US): transferrin (Tfn-FITC), phorbol myristate acetate (PMA) and ionomycin, lipopolysaccharide (LPS) from *E. coli* 0127:B8, FAC, holo-transferrin, linoleic acid, oleic acid, hemin (porcine), biliverdin-hydrochlorid, AS8351 (311), Protoporphyrin IX, Dynasore hydrate, Pitstop 2, 2,2 Bipyridyl (BIP), propidium iodid and calcein-acetoxymethyl ester (Calcein-AM) was obtained from Biozyme Scientific GmbH (Vienna, Austria). Tin Proto-porphyrin IX was from Bio-techne Ltd (Abingdon, UK). GP1Δ-Ig (Machupo virus glycoprotein) and the control protein SNIT were generated as recently described[22]. Abraxane was obtained from Celgene GmbH (Summit, US), FIX and PERM® from Nordic-MUbio (Susteren, NLD) and [methyl-3H]-thymidine from Perkin Elmer/New England Corporation (Wellesley, MA).

**Serum-free and protein-free medium**. Cells were maintained in RPMI 1640 medium, supplemented with 2 mM L-glutamine, 100 U/ml penicillin, and 100 µg/ml streptomycin without FCS. The protein-free medium was further supplemented with different HSA proteins, as mentioned in the text.

**Albumin proteins**. In this study we have used two human serum albumin proteins (HSA) which were plasma-derived from human blood: HSA (Albiomin) from Biotest (Dreieich, DE), which is has clinical grade, and HSA from Sigma-Aldrich (St. Louis, US). Fatty acid free HSA (dHSA) was purchased from Sigma-Aldrich,

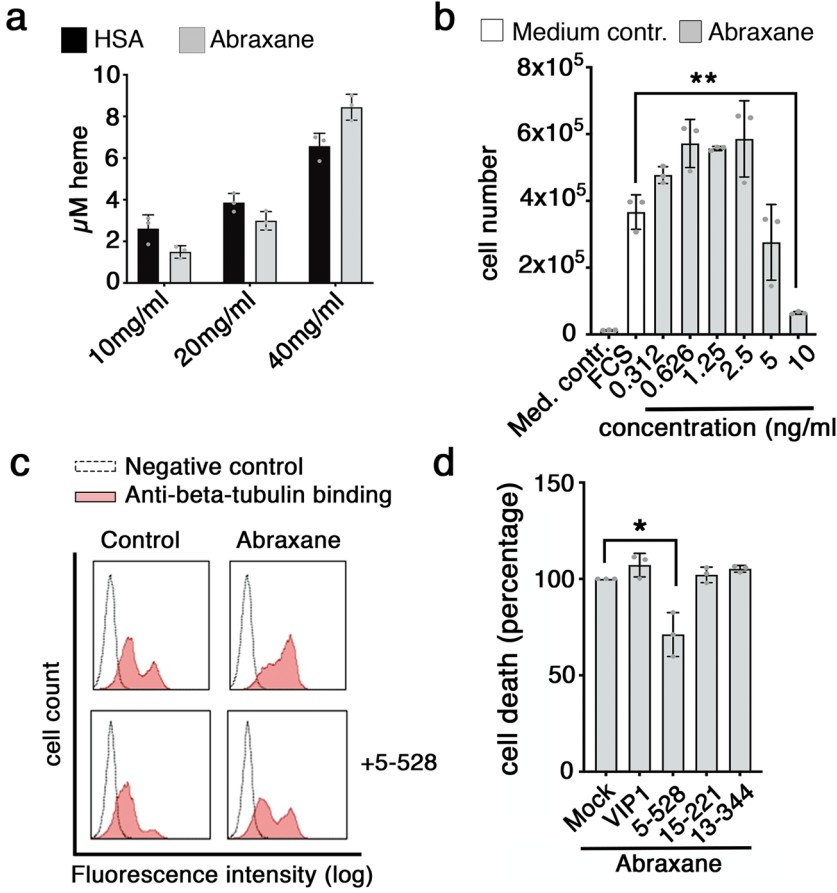

**Fig. 8 Abraxane induced cell death is executed in part via CD71. a** Bar graph illustrates the concentration of heme (µM), which was bound to HSA and HSA bound Paclitaxel particles (Abraxane) at different concentrations via a Heme Assay Kit (*n* = 3). **b** Toxic effect of Abraxane on Jurkat T cells. Cells were incubated with Abraxane for 6 days (*n* = 3). RPMI medium without serum was used as medium control. **c** Analysis of anti-beta-tubulin antibody binding to Jurkat T cells by flow cytometry. Data show mean values of MFI of anti-beta-tubulin-antibody binding to beta-tubulin in presence or absence of Abraxane (100 ng/ml), with or without mAb 5-528 (CD71). Data shown is representative of three independently performed experiments. **d** Effect of CD71 mAbs on Abraxane-treated cells. Jurkat T cells were incubated with Abraxane (100 ng/ml) in combination of CD71 mAbs (10 µg/ml), VIP1, 5-528, 15-221, or 13-344 (control). After 72 h, cell viability was analyzed by PI staining and flow cytometry (*n* = 3). **b, d** Error bars represent standard deviation (SD). *P < 0.05 and **P < 0.01 (one-way ANOVA followed by Tukey's multiple comparison test).

which was generated from HSA (Sigma-Aldrich) due to charcoal treatment. Recombinant HSA expressed in S. cerevisiae (rHSA) or in Oryza sativa (OSrHSA) was acquired from Sigma-Aldrich. BSA was purchased from GE Healthcare (Pasching, AT). The endotoxin levels in all recombinant probes was <1EU/mg.

**Cell isolation and stimulation.** Buffy coats from healthy donors were purchased either from the Austrian Red Cross or University Clinic for Blood Group Serology and Transfusion Medicine, Medical University of Vienna (both, Vienna, Austria). Peripheral blood mononuclear cells (PBMC) were isolated from heparinized buffy coats via standard density gradient centrifugation using Ficoll-Paque™ Plus (GE Healthcare, Chalfont St Giles, US). T (CD3+) cells and monocytes were purified from PBMCs using the MACS system (Miltenyi Biotec, Bergisch Gladbach, DEU). T cells (total CD3+ cells) were obtained via depletion of CD11b, CD14, CD16, CD19, CD33, and MHC class II-positive selection using biotinylated CD14 mAb. Dendritic cells (DCs) were generated by culturing purified blood monocytes for 6 days with a combination of GM-CSF (50 ng/ml) and IL-4 (35 ng/ml). On day 6 of differentiation, DCs were stimulated using 1 µg/ml LPS for 24 h. Murine splee-nocytes (WT) and TFRC knockin spleenocytes (KI) from mice expressing mutated human transferrin receptor were generated as described previously[7].

**HSA, hemin, biliverdin, and protoporphyrin preparation and conjugation.** Hemin, biliverdin, and protoporphyrin were dissolved in 1 M NaOH ($3 \times 10^{-7}$ M) and incubated with 10 mg/ml HSA for 1 h at room temperature. The protein was dialyzed for 48 h against 1× PBS, changing the buffer after 24 h, to remove non-bound hemin. HSA-heme, HSA-biliverdin and HSA-protoporphyrin were then sterile filtered. The loading of HSA with hemin was analyzed after filtration by measuring the absorbance spectra. The ratio of HSA: heme was 1: 1,5 ± 0,36 (*n* = 8). At a protein concentration of 200 µg/ml, HSA-heme ($1,8 \times 10^{18}$ protein

molecules, 251 µg/ml iron) and transferrin ($1,5 \times 10^{15}$ protein molecules, 0.3 µg/ml iron) carry $2,7 \times 10^{18}$ and $3 \times 10^{15}$ iron molecules, respectively.

HSA was labeled with Alexa Fluor 488 (NHS; HSA-AF488) dye according to the manufacturer's instructions (Thermo Fisher Scientific Inc., Waltham, US). After labeling a purification run using a S200 column (Superdex 200, 10/300 GL, GE Healthcare Life Sciences) on an ÄKTA purifier system (GE Healthcare Life Sciences, Pittsburgh, US) was performed. Protein was concentrated and set to a concentration of 12 mg/ml.

**Heme-assay kit.** To quantify the amount of hemin bound to different HSA a Heme Assay Kit protocol was used. Heme Assay Kit (MAK316) was purchased from Sigma-Aldrich (St. Louis, US). Hemin level was determined using manufacturer's protocol and calibration standards provided. In brief, HSA and Abraxane were diluted at different concentrations and applied to 96-well plates. The reaction mix (supplied with the kit) was added, and the plates were incubated for 5 min at room temperature in the dark. Changes in absorbance at 405 nm were measured using a 96-well plate reader (Bio-Rad Laboratories, Hercules, CA).

**Culture and activation of T cells.** MAXISORP Nunc-Immuno plates (Thermo Scientific, Waltham, MA) were coated overnight at 4 °C with CD3 mAb (5 µg/ml) in combination with CD28 (5 µg/ml) mAb. The plates were then washed to remove unbound mAbs and purified T cells ($2 \times 10^5$/well) were then added to the respective wells. T cell proliferation was assessed via cellular incorporation of [methyl-3H]-thymidine. Cells were labeled with 0.05 mCi/well of [methyl-3H]-thymidine on day 3 and cultured for another 18 h prior to harvesting. Detection was performed on a microplate counter (MicroBeta2; Perkin). All assays were performed in triplicates and readings are displayed as counts per minute.

**Flow cytometry.** For cell surface staining, $5 \times 10^5$ cells per staining were incubated with either unconjugated mAb, PE-conjugated or FITC-conjugated mAb for 30 min at 0 °C. For unconjugated mAbs or proteins (GP1ΔIg), secondary Oregon Green 488 (IgG) or APC-conjugated donkey anti-human IgG was added to samples as second step reagents. In some experiments, cells were pretreated with different antibodies or proteins like CD71 mAbs (5 μg/ml), transferrin (50 μg/ml), Abraxane (100 ng/ml), HSA or HSA-heme (both 200 μg/ml). $K_D$ value for HSA-heme was calculated as described[61]. Flow cytometry analyses were performed using FACS-Calibur flow cytometer (Becton Dickinson, Franklin Lakes, NJ).

Intracellular stainings were performed by fixing cells for 20 min at room temperature using FIX solution followed by permeabilization in the presence of anti-beta−tubulin-AF488 (1 μg/ml) using PERM solution for 20 min at room temperature. These cells were pretreated with Abraxane (100 ng/ml) for 2 h at 37 °C with or without 10 μg/ml mAb 5-528. 1% BSA/PBS was used as staining buffer. DNA staining was performed with propidium iodide.

**Protein concentration.** Some intracellular stainings were performed with cyto-plasmatic antibodies. Jurkat T cells were cultivated for 24 or 48 h with 10% FCS medium with or without HSA-heme or FAC (25 μg/ml). After 24 h expression of human TFR1 and after 48 h expression of human ferritin was assessed by using Alexa 647 conjugated anti-human TFR1 and Alexa 647 anti-human ferritin antibody.

**Internalization.** For endocytosis cells ($5 \times 10^5$) were incubated with different concentrations of HSA-FITC or Tfn-FITC at 0 °C or 37 °C for 1 h and were ana-lyzed by FACS. For microscopy, Jurkat T cells were exposed to 1 mg/ml HSA-AF488 for 1 h at 0 °C or 37 °C for internalization. 1% BSA/PBS was used as washing buffer. Epifluorescence microscopy was performed with an Eclipse Ti-E (Nikon) inverted microscope system with a high NA objective (×100 magnification, NA = 1.49, Nikon SR APO TIRF). The sample was illuminated by using a 488 nm diode laser (iBeam smart Toptica). Appropriate filters to visualize GFP were used. For recording the emission light we used a backside-illuminated EM-CCD camera (Andor iXon Ultra 897). Metamorph (Molecular Devices) was used to control hardware elements.

**Determination of the intracellular LIP level.** Jurkat T cells were incubated for 2 h with 10% FCS medium, HSA-heme (200 μg/ml) or FAC (25 μg/ml). At the used concentrations, the amount of iron was 251 μg/ml and 5 μg/ml for HSA-heme and FAC, respectively. Cells were collected and washed twice with 1% BSA/PBS (washing buffer) and incubated at a density of $2 \times 10^5$ cells for 15 min at 37 °C in the dark with 30 nM calcein-AM. Then, the cells were washed twice and treated with a combination of 10 μM 311 ($Fe^{3+}$ chelator) and 250 μM BIP ($Fe^{2+}$ chelator) for 1 h at RT. Afterwards, cells were analyzed by flow cytometry. The LIP level was calculated as the difference in mean fluorescence intensity of the cells treated with or without 311/BIP (Δ MFI).

**ELISA assay.** Transferrin was detected and measured by ELISA. Flat bottomed, 96 well ELISA plates (Corning Life Sciences, Tewsbury, MA) were coated with 2% blood derived HSA or 100 μg/ml transferrin overnight at 4 °C. The plate was washed twice with a PBS-Tween 0.05% solution. Blocking was performed by adding 200 μl of a 2% PBS-BSA solution to each well and incubating again over-night at 4 °C. After washing with PBS-Tween, 50 μl of mAb 13-344, directed against transferrin, was added and incubated for 2 h at 4 °C in the dark. Transferrin was used as positive control. The plate was washed then 3 times with PBS-Tween. Afterwards, the 13-344 was detected with an anti-mouse antibody labeled with alkaline phosphatase (1:2000) and incubated for 1 h at room temperature. Unbound antibodies were removed by washing 3 times. Substrate buffer containing diethanolamine was added to analyze the bound antibodies colorimetrically using an ELISA reader (Bio-Rad Laboratories, Hercules, CA). 1% NaOH was used as stop solution.

**Proliferation assay.** Jurkat T cells were cultured at $3 \times 10^4 - 1 \times 10^5$/well in 24-well plates (Costar, Sigma Aldrich) using RPMI 1640 medium supplemented with various proteins (10% FCS, HSA, BSA, HSA-heme, HSA-biliverdin, HSA-proto-porphyrin, human transferrin, heme, biliverdin) at different concentrations instead of 10% FCS or 10% FCS with FAC. In some experiments anti-human CD71 mAbs (10 μg/ml) were added. Fatty acids, linoleic acid and/or oleic acid were used at a final concentration of 100 mM. Abraxane, MiTMAB, Dynasore, Pitstop 2, EDTA, 311 or Tin Protoporphyrin were applied at different concentrations. Proliferation was monitored with light microscopy. Cells were harvested at day 6 and the cell viability was measured in triplicates with CASY cell counter (Roche Innovatis AG, Bielefeld, Germany). In other experiments cells were assessed via cellular incor-poration of [methyl-3H]-thymidine. Cells were labeled (0.05 mCi/well) of [methyl-3H]-thymidine on day 3 and cultured for another 18 h prior to harvesting. Detection was performed on a microplate scintillation counter. All assays were performed in triplicates.

**RNA isolation and qPCR.** Total cellular RNA was isolated using peqGOLD TriFast reagent (peqLab, Erlangen, DEU). For isolation, $5 \times 10^5$ cells/ml were lysed in 500 μl of TriFast reagent and isolated according to the manufacturer's protocol. 1 μg of total RNA per sample was reverse transcribed using H-Minus-Reverse Transcriptase (Thermo Fisher Scientific Inc., Waltham, US) and oligo-dT$_{18}$ pri-mers. Quantitative real-time PCR was performed with a CFX96 Real-Time PCR Detection System (Bio-Rad, Hercules, CA) using SYBR Green qPCR master mix (Quanta Biosciences, Gaithersburg, MD) for detection. Detection was performed according to the manufacturer's protocol. cDNA was amplified using a standard program (10 min at 95 °C, 40 cycles of 15 s at 95 °C/15 s at 60 °C/45 s at 72 °C).

CD3E or β2m was used as an endogenous reference gene. Primers specific for HO-1, TFR1, TF, IRP1 and ferritin were designed using Primer 3 Plus software and were custom synthesized by Sigma-Aldrich. Sequences were as follows: HO-1 fwd 5′-AAGATTGCCCAGAAAGCCCTGGAC-3′ and HO-1 rev 5′-AACTGTCGCC ACCAGAAAGCTGAG-3′; TFR1 fwd 5′- ACTTCTTCCGTGCTACTTCCAG-3′ and TFR1 rev 5′-ACTCCACTCTCATGACACGATC-3; TF fwd 5′-ATGCTGCC ACCTCTAGAATGTC-3′ and TF rev 5′-TTTGGGCTTTTGGACTCAGC-3′; IRP1 fwd 5′-TCCCTGGTGAGAATGCAGATG-3′ and IRP1 rev 5′- TCTTGCCAGTA TCCAGCTTGAC-3; ferritin fwd 5′-CAGAACTACCACCAGGACTCAG-3′ and ferritin rev 5′- GTCATCACAGTCTGGTTTCTTG-3. Data analysis was performed using CFX Manager Software (Bio-Rad).

Jurkat T cells do not produce transferrin. After 6 h and 24 h of culture in the presence of either FCS, HSA, HSA-heme or FAC, expression of transferrin mRNA was analyzed by qPCR and not detectable in the Jurkat T cells.

**Cell cycle analysis.** Jurkat T cells were harvested on day 4 of culture and were fixed in 70% ethanol at 4 °C for 30 min. After washing, cells were stained with 50 μg/ml propidium iodide in the presence of 100 μg/ml RNAse A for 45 min at room temperature. DNA content was then measured on a BD FACSCalibur flow cytometer (BD Biosciences, San Jose, CA).

**Multi-channel reporter cell line assay.** To analyze the activation of downstream signaling pathways, a multi-channel reporter cell line Jurkat E6 expressing reporter genes under control of NF-κB, NFAT, and AP-1 promoter elements was used. The promotor elements drive the expression of fluorescent proteins (eCFP, eGFP, mCherry). The reporter cell line was generated by introducing constructs encoding NF-κB-eCFP, NFAT-eGFP, and AP-1-mCherry into Jurkat E6 cells. A cell clone that was negative for fluorescent proteins in an unstimulated state and strongly upregulated eCFP, eGFP and mCherry expression upon PMA/ionomycin treat-ment was selected for further use. The reporter cells were cultivated in serum-free medium supplemented with HSA, HSA-heme or transferrin and were activated via plate bound CD3 and CD28 mAb at a concentration of 5 μg/ml. To assess the activation of the respective transcription factors, cells were harvested after 4 h, 24 h, and 48 h and expression of the receptor genes eCFP, eGFP, and mCherry were measured by flow cytometry using a LSRFortessa (flow cytometer, Becton Dickinson).

**Cytokine measurements.** Supernatants from monocyte-derived DC were collected after 24 h of LPS stimulation or from unstimulated cells. Cytokines were deter-mined using manufacturer's protocol. In brief, supernatants were prepared, the reaction mix (Multiplex Assay, Merck Millipore) was added and cytokines including IL-10, IL-12p70, IL-6, and TNF-α were measured by using Luminex 100 System (R&D Systems Inc.). All measurements were performed in triplicates.

**Statistics and reproducibility.** Statistical analysis was performed using GraphPad Prism software (La Jolla, CA). One-way analysis of variance (ANOVA) followed by Tukey's multiple comparison test or unpaired, two-tailed Student's t-test, followed by Holm-Šídák test for multiple comparisons, was performed. The P-values <0.05 were considered significant and are represented as $*P < 0.05, **P < 0.01, ***P < 0.001, ****P < 0.0001$. For all statistical analysis data from at least two biological repeats performed on separate days was used. The exact number of replicates are presented in individual figure legends. Any differences in statistical significance are indicated.

**Reporting summary.** Further information on research design is available in the Nature Research Reporting Summary linked to this article.

## Data availability

All source data underlying the graphs and charts in the main figures are available in the Supplementary Data section. The data that support the findings of this study are available from the corresponding author (J.S.) upon reasonable request.

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

## Acknowledgements
We thank Petra Waidhofer-Söllner, Petra Cejka, and Claus Wenhardt for their expert technical assistance. This work was supported by the Austrian Science Fund (FWF) grant DK W1212.

## Author contributions
J.B., V.B., M.M., A.P., J.C., S.K., M.S., M.K., performed experiments and analyzed the data. R.C.G., G.J.Z., P.S., L.Ö., A.Y., H.C., H.S. provided essential reagents and gave experimental advice. J.S. supervised the research and analyzed data. J.B. and J.S. wrote the manuscript.

## Competing interests
The authors declare no competing interests.
