## [Peer Review File · Communications Biology]

Reviewers' comments:

Reviewer #1 (Remarks to the Author):

The authors present a variety of experimental evidence that methemalbumin (HSA-heme) is taken up into several different types of cells including immune cells via the CD71 receptor, also known as the transferrin receptor 1 and shown to take up iron-transferrin complexes by endocytosis. Although not addressed directly here, in this system CD71 and transferrin recycle back to the cell surface and the ferric iron is reduced to ferrous by a reductase, STEAP3.

Using almost a dozen human and mouse cells (including erythroblasts, human monocytic, epithelial and T-lymphocytes), the authors have investigated the effects of methemalbumin by stimulating cell growth and the need for functional CD71 in this process. The hypothesis tested is that heme from methemalbumin will be used as an additional or alternative iron source delivered to cells via CD71, whose surface expression is upregulated upon cell activation and growth. Heme oxygenases release iron from heme, which is then utilized for cell growth. Certain signaling events are also investigated. Addition of hemin alone did not induce proliferation suggesting that the receptor-mediated pathways provides key targeting or additional signals (e.g. c-Abl kinase H.Cao et al PMID 27226592), details of such mechanisms were not mentioned or discussed. Presumably, heme oxygenases were induced by hemin alone.

These observations are novel and would be expected to be of interest to others in the scientific community; however, further evidence is needed to strengthen the conclusions of the authors (see comments and suggestions below).

The statistical analyses used appear to be appropriate and sound. The data presentation needs improvement especially in Figures 1 and 3. The heat maps are not that easy to follow (for this reviewer).

In spite of the large amount of data and experimental approaches, there are several weaknesses with this study.

1. The authors need to characterize their methemalbumin complex. No information is provided to establish the stoichiometry of heme: HSA. Is the complex 1:1 or 2:1? Absorbance spectra are required.
2. The literature support two binding sites for heme on HSA of differing affinities. Also, heme binds where fatty acids would reside. Would fatty acids in the plasma displace heme?
3. Is the concentration and saturation of heme-HSA the same before and after sterile filtration?
4. Is the HSA used endotoxin free?
5. Heme-albumin has some peroxidase activity that might affect its interaction with cells (Kamal & Behere J Inorg Chem 2002 7:273). Would the authors please address this.
6. While their observations are consistent with endocytosis of heme-HSA by CD71, and the data have been interpreted as such; they do not prove internalization of the heme-albumin into vesicles of the endocytosis pathway e.g. coated pits or endosomes. Also, there are no data to show albumin recycling. This is also needed because it is an important part of the established CD71 cycle. Such data are needed in addition to the experiments using cells expressing CD71 that does not "internalize". Perhaps label the HSA with AlexaFlour and monitor heme-HSA internalization with immunofluorescence microscopy?
7. The role of albumin to maintain colloidal osmotic pressure is established. The authors use the term "protein free" medium? What is meant by this? How is the colloidal osmotic pressure of the "protein-free" medium maintained? Also, serum-free medium generates oxidative stress in cultured cells. Were any parameters of oxidative stress determined?
8. Given the unique and localized interactions between protein ligands and their specific receptors, it

is somewhat surprising that Abraxane, an HSA-based chemotherapeutic packed as HSA nano-particles also appears to be taken up by CD71 because antibodies to this receptor reduce its toxic effects. Nevertheless, the authors' data support different binding sites for these various ligands, thus CD71 function in terms of ligand binding is reminiscent of scavenger receptors.

9. DNA synthesis requires ribonucleotide reductase that has an obligatory requirement for iron. This information needs to be included in the manuscript.

10. Introduction. The role for albumin for heme clearance is perhaps overstated. Albumin has been described for decades as a "reservoir" for heme because it is seen only when haptoglobin and hemopexin are depleted (Muller-Eberhard, Javid, Liem et al. Blood(1968) 32:811) The capacity of the hemopexin system needs to be updated to account for its extensive recycling. Furthermore, in many clinical situations the heme binding site on albumin is not available because it is occupied by drugs. In addition, most receptor-mediated uptake systems are very rapid and heme-albumin is not rapidly cleared from the circulation. It would be interesting if the authors would address in the manuscript the following points. The potential capacity of the clearance system they are proposing including the fact that it is established that transferrin receptors are rapidly down regulated when iron levels rise and that heme-albumin is not rapidly cleared from the plasma. Also, there may be differences between the function of CD71 in untransformed and transformed cells.

11. Page 4 Re "HO-1 degrades heme into biliverdin which is then further degraded into bilirubin and other down-stream products including NO and CO 19, 21, 22". Heme oxygenases (in the SER) and biliverdin reductases (in cytosol) are both needed to degrade heme to bilirubin and CO. Would the authors please check, I am not aware that heme degradation is also a source of NO as stated – if so, please edit the sentence to reflect the different sources of NO and CO.

12. Albumin is a plasma protein, why do the authors use the term "Blood-derived" HSA? Please explain or change.

13. Fig. 1. This data presentation is poor and very hard to follow. Please edit.

14. Fig 3. Using a concentration of $\mu\text{g/ml}$ rather than molarity is confusing. Are the amounts of iron equivalent in the HSA-heme and iron-transferrin?

15. In experiments with serum, given the avidity of hemopexin for heme (K_d less than 1pM) and its high levels in serum, it is surprising that the heme is not transferred to hemopexin from heme-albumin. Would the authors please address this.

16. The authors state "given the abundance of HSA, reveals a significant amount of iron handled by HSA, which might provide sufficient iron for cells in the absence of transferrin." Which conditions or diseases are the authors referring to? This information needs to be included.

Reviewer #2 (Remarks to the Author):

Dear Authors,

Thank you for the opportunity to review your manuscript. I have listed several issues which need to be addressed. There are listed in order in which they appear in the manuscript.

-The introduction states that "many studies have shown that defects of the receptor are more severe than defects or mutations affecting transferrin" but only one reference is given.

-Please show data in Supplementary Figure 1b, Supplementary Figure 4, and Figure 4a as bar graphs,

not heat maps.

-The difference in cell number between HSA- and HSA-heme-treated cells in Supplementary Figure 2 is mild. TF1 cells appear to proliferate more with HSA treatment than with HSA-heme treatment! This suggests that the effect of HSA-heme on proliferation is cell line-dependent. Please comment on this.

-Please specify clearly which cell lines used in this manuscript are primary and which are immortalized or cancer-derived. This is important information to consider when interpreting the data.

-For the antibodies generated by the authors, please provide references or data on their validation for use against their intended antigens. What epitopes do these antibodies recognize?

-Do any of the cell lines used in this study synthesize transferrin? Even though serum-free media was employed by the authors, endogenous transferrin production by the cell lines would complicate the interpretation of results. Please analyze conditioned media for presence of transferrin.

-An additional control that would strengthen the manuscript is treatment of cells with protoporphyrin IX.

-The interpretation of the results in Fig. 2b is flawed. The authors state that "murine splenocytes expressing mutated, human CD71 receptors, which cannot internalize their cargo failed to proliferate in response to HSA-heme". The wild-type splenocytes also fail to proliferate in response to HSA-heme!

-To what regions of transferrin receptor does HSA-heme bind? This can be inferred based upon the data presented in this manuscript.

-There's no legend for Fig. 2f in the figure itself-- what is the significance of the white and gray curves?

-The text legend for Supplementary Figure 7 is entitled "CD71 receptor expression on EBV cell lines expressing". Please correct.

-The authors state that YK01 and OTHAKA cells differ in heme oxygenase 1 levels. Are there any other differences in these cell lines that could explain the difference in their response to HSA-heme treatment?

-The figures can be presented more clearly. For example, the y-axis on Fig. 3b is labeled only as 'fold induction'. The y-axis on Fig. 3e is labeled only as 'pg/mL'.

-The results in Fig. 3c are not compelling, as they indicate that both HSA and HSA-heme upregulate gene expression in similar fashion. The changes in gene expression in HSA-heme-treated cells can be attributed to HSA, not heme. Also, why did the authors focus solely on NF- κ B, AP1, and NFAT as read-outs for gene expression? Can the authors provide literature references indicating that these genes are regulated by degradation products of heme?

-The data in Fig. 3d is too small to evaluate. It would be easier if the curves were quantified as well.

-To demonstrate that HSA-heme uptake is a valid route of cellular iron import, the authors need to evaluate their cell lines for changes in iron-dependent gene and protein expression and in iron and heme levels. If transferrin receptor is truly a receptor for HSA-heme, HSA-heme binding to transferrin receptor must have some physiologic impact on cellular heme and iron physiology. The authors need to investigate what impact HSA-heme binding to transferrin receptor has on cell physiology in much more detail. This is a major issue that needs to be addressed. Also, is it possible that HSA-heme binding to transferrin receptor is a mechanism for heme import into the cell, not for iron import into the cell?

-Please provide statistical analysis for data in Fig. 4b.

-Use of anti-transferrin receptor antibody prevents abraxane-induced cell death by only 25%. This suggests that transferrin receptor plays a minor role in abraxane-induced toxicity. Please comment.

-How do the authors think that HSA-heme binding to transferrin receptor results in HSA-heme import into cells? Is this through endocytosis of the complex? None of the experiments address the mechanism by which HSA-heme binding to transferrin receptor leads to cellular import of HSA-heme. This is a major issue that needs to be addressed.

-The discussion is too brief. Are there disease conditions in which albumin and transferrin levels are low? In these conditions, is iron homeostasis impaired by inadequate cellular iron import? The Andrews lab published reports several years ago indicating that transferrin receptor has non-canonical roles-- how do the results in this manuscript align with the concept of non-canonical roles for transferrin receptor?

-The discussion states that "the high amounts of HSA molecules guarantee an efficient buffer system of toxic and HSA binds about 30 % more free heme than hemopexin". What is a 'buffer system of toxic'?

My main concerns are that the authors have not substantially shown that HSA-heme binding to transferrin receptor impacts cellular heme and iron homeostasis (by showing expected changes in gene and protein expression and in heme and iron levels at the very least) nor have they shown the mechanism by which HSA-heme binding to transferrin receptor leads to cellular heme import.

Reviewer #3 (Remarks to the Author):

This work presents evidence that a conjugate of heme with human serum albumin (HSA) stimulates proliferation of Jurkat and other human cell lines, or murine cell lines transfected with wild type human transferrin receptor 1 (TfR1). The authors conclude that heme-BSA is internalized into cells via specific binding to TfR1. They further show that the stimulating effects of heme-BSA on cell proliferation are abolished in a lymphoblastoid cell line from a patient with heme oxygenase 1 (HO-1) deficiency, and conclude that HO-1 activity is essential for these responses. The overall data are potentially interesting and the main conclusion is provocative. However, there are several important issues that reduce enthusiasm.

1) The major problem of this work is that the only readouts in the experiments are cell proliferation or signaling. To support their claims, the authors should demonstrate that heme-BSA results in cellular iron loading. This can be done by various assays, for instance by measuring ferritin expression, IRP1 and IRP2 activities, IRP2 expression, or levels of labile iron. It will also be important to determine whether heme-BSA promotes TfR1 mRNA degradation, a known response to iron loading. Can the heme-BSA-mediated cell proliferation be blocked by iron-chelating drugs?

2) It is not clear why free heme does not stimulate cell proliferation (suppl. Fig. 3). Heme is known to easily permeate cells and release iron following its degradation by HO-1, which in turn modulates cellular iron metabolism (for instance, see PMID 1992460). The effects of free heme on cell iron status should be measured as discussed above.

3) To support the claim that TfR1 operates as a receptor for heme-BSA, the authors should perform direct binding assays and determine the affinity constant.

Minor issues:

1) Did BSA used in Fig. 1c contain heme?

2) How much was the expression of transfected human TfR1 (wild type or mutant) compared to endogenous in the murine cells? Did transfected human TfR1 retain the iron responsive elements (IREs) in its mRNA?

3) Do HO-1 inhibitors (such as tin protoporphyrin IX) block the stimulating effects of heme-BSA in Jurkat cells and the other human cell lines?

4) Biliverdin is not “a protoporphyrin without iron”, it is a degradation product of heme, without intact protoporphyrin ring.

5) NO is not a downstream product of the HO-1 reaction; the only downstream gaseous product is CO.

6) The first sentence of the introduction is not absolutely correct. While iron is essential for the vast majority of known forms of life, there are some minor exceptions such as lactobacilli or *Borrelia burgdorferi* (see PMID 9269745 and 10834845).

Response to Reviewers

Reviewer #1:

We thank the reviewer for the careful and critical evaluation of our work and the valuable suggestions. We have now restructured and revised large parts of the manuscript according to the proposals and comments of the reviewers. We also discuss the potential impact of receptor-mediated signal pathways (e.g. c-Abl kinase) in the proliferation response of cell induced by HSA-heme (page 8, line 25).

To the points raised we would like to respond as follows:

Reviewer: *The authors need to characterize their methemalbumin complex. No information is provided to establish the stoichiometry of heme: HSA. Is the complex 1:1 or 2:1? Absorbance spectra are required.*

Response: We have now analyzed the absorbance spectra of HSA upon loading with heme and observed that the ratio of albumin : heme is 1 : 1,5 ± 0,36 (n=8). This information is now also given in the results section (page 11, line 23).

2. The literature support two binding sites for heme on HSA of differing affinities. Also, heme binds where fatty acids would reside. Would fatty acids in the plasma displace heme?

The literature supports at least two binding sites for heme on HSA of differing affinities. One site of HSA binds heme with high affinity and this binding is not affected by fatty acids (Grinberg et al. Free Radical Biology & Medicine, 1999, 26:214). The other site of HSA bind heme with low affinity and is used by various hydrophobic molecules, including fatty acids. Yet, it is intriguing that binding of other natural ligands or drugs may influence the low-affinity interaction of heme with HSA. This information is now also presented in the text on page 3, line 17-22.

3. Is the concentration and saturation of heme-HSA the same before and after sterile filtration?

The loading of HSA with heme was determined after sterile filtration (page 11, line 22).

4. Is the HSA used endotoxin free?

We used in this study plasma-derived or recombinant HSA. Plasma-derived HSA is a medicament (Albion) and the endotoxin levels in all recombinant proteins was < 1EU/mg

according to the product specification. This information is now given on page 10, line 28 – page 11, line 5.

5. Heme-albumin has some peroxidase activity that might affect its interaction with cells (Kamal & Behere J Inorg Chem 2002 7:273). Would the authors please address this.

We discuss the role of the peroxidase activity of heme-albumin in the new version of our manuscript on page 8, line 26 – page 9, line 2.

6. While their observations are consistent with endocytosis of heme-HSA by CD71, and the data have been interpreted as such; they do not prove internalization of the heme-albumin into vesicles of the endocytosis pathway e.g. coated pits or endosomes. Also, there are no data to show albumin recycling. This is also needed because it is an important part of the established CD71 cycle. Such data are needed in addition to the experiments using cells expressing CD71 that does not “internalize”. Perhaps label the HSA with AlexaFlour and monitor heme-HSA internalization with immunofluorescence microscopy?

We have addressed this important point in more detail and show in our new Figure 3e the internalization of heme-HSA in comparison with transferrin. This information is now also given in the results section (page 5, line 16-19) and in the discussion (page , line). In addition, we have now also used inhibitors of clathrin-mediated endocytosis (MitMAB, Dynasore, Pitstop 2) and observed that heme-HSA induced proliferation is inhibited in the presence of these clathrin and dynamin inhibitors (new Figure 3g, and in the text page 5, line 16-19). We have not addressed the question if albumin is recycling upon CD71-mediated endocytosis in this study. Since albumin is the only protein/amino acid source for the cells in our protein-free system, we assume a strong bias and need in the cells to degrade and use albumin as nutrient.

7. The role of albumin to maintain colloidal osmotic pressure is established. The authors use the term “protein free” medium? What is meant by this? How is the colloidal osmotic pressure of the “protein-free” medium maintained? Also, serum-free medium generates oxidative stress in cultured cells. Were any parameters of oxidative stress determined?

“Protein free medium” means that we have no other proteins in the medium except HSA or HSA-heme if not otherwise indicated. HO-1 expression is an indicator of oxidative stress. The results presented in our Figure 4b demonstrate that HO-1 expression is not upregulated in the medium control compared to medium supplemented with FCS or albumin.

8. *Given the unique and localized interactions between protein ligands and their specific receptors, it is somewhat surprising that Abraxane, an HSA-based chemotherapeutic packed as HSA nano-particles also appears to be taken up by CD71 because antibodies to this receptor reduce its toxic effects. Nevertheless, the authors' data support different binding sites for these various ligands, thus CD71 function in terms of ligand binding is reminiscent of scavenger receptors.*

Human CD71 is indeed a promiscuous cell entry carrier. Its primary function is the import of iron but several other ligands including ferritin, arenaviruses or malaria parasite use CD71 to enter cells. So, CD71 is not only the entry site for proteins but also for large molecular complexes. Our study demonstrates that HSA-heme is another factor that utilizes this special cell entrance. The 3 CD71 mAbs used in this study recognize 3 different epitopes. This information is now provided in our new Supplementary Figure 9 and 11. The binding sites of mAb VIP-1 and 5-528 are overlapping (page 5, line 6-8). Both mAbs inhibit binding of HSA-heme to CD71, which is presented in our new Figure 3a and 3b, where 5-528 is less efficient. However, mAb VIP-1 inhibits cell proliferation, whereas mAb 5-528 does not. Thus, the inhibitory VIP-1 is not suitable to revert the cell toxic effect of Abraxane, whereas 5-528 is able to partly revert the Abraxane killing because the mAb is not affecting the proliferation and viability of the cells *per se*.

MAb 15-221 recognizes a different epitope. MAb 15-221 inhibits binding of GP-1 protein from Machupo Virus but does not interfere with HSA-heme binding.

This information is now also given in the discussion (page 9, line 7-14).

9. *DNA synthesis requires ribonucleotide reductase that has an obligatory requirement for iron. This information needs to be included in the manuscript.*

This information is now included in the Introduction of our manuscript (page 3, line 3).

10. *Introduction. The role for albumin for heme clearance is perhaps overstated. Albumin HSA been described for decades as a "reservoir" for heme because it is seen only when haptoglobin and hemopexin are depleted (Muller-Eberhard, Javid, Liem et al. Blood(1968) 32:811) The capacity of the hemopexin system needs to be updated to account for its extensive recycling. Furthermore, in many clinical situations the heme binding site on albumin is not available because it is occupied by drugs. In addition, most receptor-mediated uptake systems are very rapid and heme-albumin is not rapidly cleared from the circulation. It would be*

interesting if the authors would address in the manuscript the following points. The potential capacity of the clearance system they are proposing including the fact that it is established that transferrin receptors are rapidly down regulated when iron levels rise and that heme-albumin is not rapidly cleared from the plasma. Also, there may be differences between the function of CD71 in untransformed and transformed cells.

We have now dampened the role of albumin in the clearance of heme (page 3, line 17).

The function of CD71 in cell biology is certainly more complex than its key role in iron uptake. Already the expression of CD71 HSA has been reported to induce and regulate signaling in cells. Engagement of CD71 induces signaling responses in cells and non-canonical functions have been described for CD71. To analyze the impact of HSA-heme binding to CD71 in all facets of its functional repertoire is certainly very interesting and important but is beyond of the scope of our study. In addition, it is of course needed to study the interplay of HSA-heme and other heme-binding molecules including hemopexin to better understand its position in the process of heme-clearance. This information is now given in the discussion (page 7, line 18-29).

11. Page 4 Re “HO-1 degrades heme into biliverdin which is then further degraded into bilirubin and other down-stream products including NO and CO 19, 21, 22”. Heme oxygenases (in the SER) and biliverdin reductases (in cytosol) are both needed to degrade heme to bilirubin and CO. Would the authors please check, I am not aware that heme degradation is also a source of NO as stated – if so, please edit the sentence to reflect the different sources of NO and CO.

We are sorry for the mistake. Heme degradation is not a source of NO. The sentence on page 6, line 8 has been corrected in the revised version of the paper.

12. Albumin is a plasma protein, why do the authors use the term “Blood-derived” HSA? Please explain or change.

We now use “plasma-derived” instead of “blood-derived” in the context of albumin.

13. Fig. 1. This data presentation is poor and very hard to follow. Please edit.

The data presentation in Figure 1 has been revised and we hope that it is now easier to follow.

14. Fig 3. Using a concentration of $\mu\text{g/ml}$ rather than molarity is confusing. Are the amounts of iron equivalent in the HSA-heme and iron-transferrin?

Thank you to ask this important question. It was really interesting to see that addition of HSA-heme was equally potent to promote proliferation of cells like holo-transferrin (Figure 1e), indicating that both proteins provide sufficient iron (and amino acids) for the cells to grow and expand. We have now measured and calculated the amounts of iron loaded onto our albumin compared to transferrin. At a protein concentration of 200 $\mu\text{g/ml}$, HSA-heme ($1,8 \times 10^{18}$ protein molecules) and transferrin ($1,5 \times 10^{15}$ protein molecules) carry $2,7 \times 10^{18}$ and 3×10^{15} iron molecules, respectively. This information is now also presented in the text on page 11, line .

15. In experiments with serum, given the avidity of hemopexin for heme (K_d less than 1pM) and its high levels in serum, it is surprising that the heme is not transferred to hemopexin from heme-albumin. Would the authors please address this.

It is indeed surprising that although hemopexin is the major ligand for heme a part of heme is still bound to HSA. We have no answer for this paradox. In order to avoid the complex situation, present in serum/plasma, we have chosen to study the effect of HSA-heme in a serum/plasma and protein-free environment. This is of course an artificial or even more artificial condition than 10 % FCS but suitable to elucidate the single effect of HSA-heme without the contribution or disturbances of other factors.

16. The authors state “given the abundance of HSA, reveals a significant amount of iron handled by HSA, which might provide sufficient iron for cells in the absence of transferrin. “ Which conditions or diseases are the authors referring to? This information needs to be included.

We have changed this statement in the Introduction and discuss the potential biological role of HSA-heme in more detail in the new version of our paper (page 3, line 26-27).

Response to Reviewer

Reviewer #2:

We thank the reviewer for the critical evaluation of our work and the valuable suggestions. We have now restructured and revised large parts of the manuscript according to the proposals and comments of the reviewers. To the points raised we would like to respond as follows:

Reviewer -*The introduction states that "many studies have shown that defects of the receptor are more severe than defects or mutations affecting transferrin" but only one reference is given.*

Response: We have now included more references concerning to support this statement (page 3, line 11).

-Please show data in Supplementary Figure 1b, Supplementary Figure 4, and Figure 4a as bar graphs, not heat maps.

We have now changed the respective figures from heat maps to bar graphs.

-The difference in cell number between HSA- and HSA-heme-treated cells in Supplementary Figure 2 is mild. TF1 cells appear to proliferate more with HSA treatment than with HSA-heme treatment! This suggests that the effect of HSA-heme on proliferation is cell line-dependent. Please comment on this.

We have tested the impact of HSA-heme on different cell lines. In comparison with plasma-derived HSA we have observed that cells need less HSA-heme to proliferate. This is also the case for TF1 cells. TF1 cells are erythroid cells which start to differentiate in response to heme towards erythrocyte-lineage. Such a differentiation process is likely to be accompanied with a reduced proliferation rate in TF1 cells. The role of HSA-heme on the proliferative response of other cell types needs to be tested in more detail. This is now also mentioned in the text on page 7, line 18-29.

-Please specify clearly which cell lines used in this manuscript are primary and which are immortalized or cancer-derived. This is important information to consider when interpreting the data.

This information is now given in the Methods section on page 9, line 19-27.

For the antibodies generated by the authors, please provide references or data on their validation for use against their intended antigens. What epitopes do these antibodies recognize?

The 3 CD71 mAbs used in this study recognize 3 different epitopes. This information is now provided in our new Supplementary Figure 9 and 11. The binding sites of mAb VIP-1 and 5-528 are overlapping (page 5, line 6-8). Both mAbs inhibit binding of HSA-heme to CD71, which is presented in our new Figure 3a and 3b, where 5-528 is less efficient. However, mAb VIP-1 inhibits cell proliferation, whereas mAb 5-528 does not. This information is now also given in the discussion (page 9, line 7-14).

-Do any of the cell lines used in this study synthesize transferrin? Even though serum-free media was employed by the authors, endogenous transferrin production by the cell lines would complicate the interpretation of results. Please analyze conditioned media for presence of transferrin.

We have done most of the experiments in Jurkat T cells. These cells do not produce transferrin as analyzed by qPCR and ELISA. This information is now given in the Methods section (page 9, line 21-22).

-An additional control that would strengthen the manuscript is treatment of cells with protoporphyrin IX.

We have performed this additional control and demonstrate in our new Supplementary Figure 3 that loading of HSA with protoporphyrin IX does not promote proliferation of cells. This information is now also given in the text (page 4, line 7).

-The interpretation of the results in Fig. 2b is flawed. The authors state that "murine splenocytes expressing mutated, human CD71 receptors, which cannot internalize their cargo failed to proliferate in response to HSA-heme". The wild-type splenocytes also fail to proliferate in response to HSA-heme!

We provide in the new version of our manuscript a more detailed interpretation of the results presented in Figure 2, also in the context with the other additional data concerning HSA-heme uptake and specificity of the reaction (page 5, line 16-19).

Wild type, murine splenocytes, used in this study, do not express human CD71. Murine splenocytes (CD71-mut) were transfected with a mutated, non-functional version of human CD71.

-To what regions of transferrin receptor does HSA-heme bind? This can be inferred based upon the data presented in this manuscript.

Human CD71 is indeed a promiscuous cell entry carrier. Its primary function is the import of iron but several other ligands including ferritin, arenaviruses or malaria parasite use CD71 to enter cells. Our study demonstrates that HSA-heme is another factor that utilizes this special cell entrance. Based on our binding studies, we conclude that the binding site of HSA-heme is in proximity to the transferrin binding site (lateral part of the ectodomain of CD71) and distinct from the ferritin/pathogen contact region (apical part of the ectodomain of CD71). This information is now also given in the discussion (page 7, line 1-16).

-There's no legend for Fig. 2f in the figure itself-- what is the significance of the white and gray curves?

Fig. 2f (old version) is Figure 3c in the new version. The figure shows that Jurkat cells which do not express CD71 on the cell surface due to down-modulation by mAb VIP-1 treatment and cultured by addition of exogenous iron in form of ferric ammonium citrate (FAC), do not bind HSA-heme. A better description in the legend to the figure in the Methods section (page 9, line 24-27) is now provided.

-The text legend for Supplementary Figure 7 is entitled "CD71 receptor expression on EBV cell lines expressing". Please correct.

The title has been corrected (new Supplementary Figure 13).

-The authors state that YK01 and OTHAKA cells differ in heme oxygenase 1 levels. Are there any other differences in these cell lines that could explain the difference in their response to HSA-heme treatment?

YK01 is an EBV-transformed lymphoblastoid B cell line from a patient suffering with HO-1 deficiency. OTHAKA is an EBV-transformed lymphoblastoid B cell line generated from a healthy donor. You are right that these 2 different cell lines may nevertheless have additional molecular differences. This has not been analyzed so far. Yet, CD71 and iron-uptake are

obviously intact since YK01 cells proliferate in response to transferrin containing medium (FCS). It is intriguing that there are also other molecular differences between both cell lines. However, we provide now more evidence that HO-1 is important for HSA-heme induced proliferation. Results presented in our new Figure 5a demonstrate that proliferation of Jurkat cells in the presence of HSA-heme but not FCS is inhibited by Tin Protoporphyrin, an inhibitor of HO-1. This information is now also in the paper on page 4, line 4-8.

-The figures can be presented more clearly. For example, the y-axis on Fig. 3b is labeled only as 'fold induction'. The y-axis on Fig. 3e is labeled only as 'pg/mL'.

The description and labeling of the figures have been revised and improved.

-The results in Fig. 3c are not compelling, as they indicate that both HSA and HSA-heme upregulate gene expression in similar fashion. The changes in gene expression in HSA-heme-treated cells can be attributed to HSA, not heme. Also, why did the authors focus solely on NF-kB, AP1, and NFAT as read-outs for gene expression? Can the authors provide literature references indicating that these genes are regulated by degradation products of heme?

We wanted to analyze the potential influence of heme-labeling on signaling processes in cells. We found indeed no difference between HSA and HSA-heme at the concentrations applied in this study. It will be of course interesting to extend these studies on other signaling routes and molecules such as c-Abl kinase (page 8, line 21-25).

We are sorry but NF-kB, AP1 and NFAT have been selected because they are all important transcription factors and the read-out was the activation of these factors in a reporter-cell line. We have not analyzed the gene expression of these 3 signaling molecules.

-The data in Fig. 3d is too small to evaluate. It would be easier if the curves were quantified as well.

We have reformatted Figure 6b (old Fig. 3d) so that it is easier to evaluate. Overlay histograms are a standard format to illustrate a representative expression profile of cell surface markers on immune cells from flow cytometry data.

-To demonstrate that HSA-heme uptake is a valid route of cellular iron import, the authors need to evaluate their cell lines for changes in iron-dependent gene and protein expression and in iron and heme levels. If transferrin receptor is truly a receptor for HSA-heme, HSA-

heme binding to transferrin receptor must have some physiologic impact on cellular heme and iron physiology. The authors need to investigate what impact HSA-heme binding to transferrin receptor HSA on cell physiology in much more detail. This is a major issue that needs to be addressed. Also, is it possible that HSA-heme binding to transferrin receptor is a mechanism for heme import into the cell, not for iron import into the cell?

We have shown in various experiments in this paper that HSA-heme has a significant impact on the physiology of cells e.g. cells get enough iron to proliferate. In order to provide more insights and details for this effect, we have performed now a number of additional experiments. We have analyzed the expression of iron-dependent genes including CD71 and show in our new Figure 4c that *CD71* and *IRP1* are down-regulated in the presence of HSA-heme, whereas ferritin was not regulated. We now also demonstrate that HSA-heme-mediated cell proliferation can be blocked with iron-chelator 311, which is cell membrane permeable, but not with EDTA, which acts extracellularly (Figure 5b).

We also demonstrate in Figure 6b and 6c that HSA-heme has physiological consequences on cell function i.e. DCs differentiation and cytokine production. Thus, HSA-heme uptake is not only a mechanism to bring iron into the cell but also heme and the required amino acids (from the albumin) which the cells need to proliferate and/or to produce cytokines. The need for HO-1 activity to see HSA-heme induced proliferation further underlines that heme-degradation occurs. To further support the role of HO-1 activity, we have used protoporphyrin IX to block HO-1. The results presented in our new Figure 5a demonstrate that the drug inhibits the HSA-heme mediated proliferation but not proliferation in the presence of FCS (control). This information is now also given in the text (page 5, line 28-30 – page 6, line 1-6).

-Please provide statistical analysis for data in Fig. 4b.

We provide statistical analysis in Figure 7b (old Fig. 4b).

-Use of anti-transferrin receptor antibody prevents abraxane-induced cell death by only 25%. This suggests that transferrin receptor plays a minor role in abraxane-induced toxicity. Please comment.

It was not our intention to present it as major effect. Our selection of abraxane as a model albumin reagent was not really “lucky” because it turned out that the toxic effect of the paclitacel/HSA complex with CD71 mAbs, which are often *per se* anti-proliferative and at the end trigger cell death. The 3 CD71 mAbs used in this study recognize 3 different epitopes.

This information is now provided in our new Supplementary Figure 9 and 11. The binding sites of mAb VIP-1 and 5-528 are clearly overlapping. Both mAbs inhibit binding of HSA-heme to CD71, which is presented in our new Figure 3a and 3b, where 5-528 is less efficient. However, mAb VIP-1 inhibits cell proliferation, whereas mAb 5-528 does not. Thus, the inhibitory VIP-1 is not suitable (and does not do it) to revert the cell toxic effect of Abraxane, whereas 5-528 is able to partly revert the Abraxane killing because the mAb is not affecting the proliferation and viability of the cells *per se*. The role of CD71 in the function of Abraxane is discussed in the new version of our paper on (page 9, line 7-14).

-How do the authors think that HSA-heme binding to transferrin receptor results in HSA-heme import into cells? Is this through endocytosis of the complex? None of the experiments address the mechanism by which HSA-heme binding to transferrin receptor leads to cellular import of HSA-heme. This is a major issue that needs to be addressed.

We have addressed this important point in more detail and show in our new Figure 3e the internalization of heme-HSA in comparison with transferrin. This information is now also given in the results section (page 5, line 16-19) and in the discussion (page 7, line 18-29). In addition, we have now also used inhibitors of clathrin-mediated endocytosis (MitMAB, Dynasore, Pitstop 2) and observed that heme-HSA induced proliferation is inhibited in the presence of these clathrin and dynamin inhibitors (new Figure 3g, and in the text page , line). We have not addressed the question if albumin is recycling upon CD71-mediated endocytosis in this study. Since albumin is the only protein/amino acid source for the cells in our protein-free system albumin, there is obviously a strong bias and need in the cells to degrade and use albumin as nutrient.

-The discussion is too brief. Are there disease conditions in which albumin and transferrin levels are low? In these conditions, is iron homeostasis impaired by inadequate cellular iron import? The Andrews lab published reports several years ago indicating that transferrin receptor HSA non-canonical roles-- how do the results in this manuscript align with the concept of non-canonical roles for transferrin receptor?

We discuss the potential biological relevance of albumin/CD71 mediated iron uptake in more detail and also a link of our findings with the reported non-canonical roles of CD71 and the contribution of HSA-heme (page 8, line 18-28).

-The discussion states that "the high amounts of HSA molecules guarantee an efficient buffer

system of toxic and HSA binds about 30 % more free heme than hemopexin". What is a 'buffer system of toxic'?

We are sorry for the mistake. The sentence is now corrected (page 7, line 14).

Reviewer: *My main concerns are that the authors have not substantially shown that HSA-heme binding to transferrin receptor impacts cellular heme and iron homeostasis (by showing expected changes in gene and protein expression and in heme and iron levels at the very least) nor have they shown the mechanism by which HSA-heme binding to transferrin receptor leads to cellular heme import.*

We have analyzed the expression of iron-dependent genes including CD71 and show in our no

new Figure 4c that *CD71* and *IRP1* are down-regulated in the presence of HSA-heme, whereas ferritin was not regulated (page 5, line 28-30 – page 6, line 1-6).

We have addressed this important point in more detail and show in our new Figure 3e the internalization of heme-HSA in comparison with transferrin and a representative picture of ingested HSA-heme (Fig. 3f). In addition, we have now also used inhibitors of clathrin-mediated endocytosis (MitMAB, Dynasore, Pitstop 2) and observed that heme-HSA induced proliferation is inhibited in the presence of these clathrin and dynamin inhibitors (new Figure 3g, and in the text page 5, line 16-19). We have not addressed the question if albumin is recycling upon CD71-mediated endocytosis in this study. Since albumin is the only protein/amino acid source for the cells in our protein-free system, there is obviously a strong bias and need of the cells to degrade and use albumin as nutrient.

Response to Reviewer

Reviewer #3:

We thank the reviewer for the careful evaluation of our work and the valuable suggestions. We have now restructured and revised large parts of the manuscript according to the proposals and comments of the reviewers. To the points raised we would like to respond as follows:

Reviewer: *This work presents evidence that a conjugate of heme with human serum albumin (HSA) stimulates proliferation of Jurkat and other human cell lines, or murine cell lines transfected with wild type human transferrin receptor 1 (TfR1). The authors conclude that heme-BSA is internalized into cells via specific binding to TfR1. They further show that the stimulating effects of heme-BSA on cell proliferation are abolished in a lymphoblastoid cell line from a patient with heme oxygenase 1 (HO-1) deficiency, and conclude that HO-1 activity is essential for these responses. The overall data are potentially interesting and the main conclusion is provocative. However, there are several important issues that reduce enthusiasm.*

1) The major problem of this work is that the only readouts in the experiments are cell proliferation or signaling. To support their claims, the authors should demonstrate that heme-BSA results in cellular iron loading. This can be done by various assays, for instance by measuring ferritin expression, IRP1 and IRP2 activities, IRP2 expression, or levels of labile iron. It will also be important to determine whether heme-BSA promotes TfR1 mRNA degradation, a known response to iron loading. Can the heme-BSA-mediated cell proliferation be blocked by iron-chelating drugs?

Response: We have analyzed the expression of iron-dependent genes including CD71 and show in our now Figure 4c that *CD71* and *IRP1* are down-regulated in the presence of HSA-heme, whereas ferritin was not regulated. We demonstrate in Figure 6b and 6c that HSA-heme HSA physiological consequences on cell function i.e. DCs differentiation and cytokine production. Thus, HSA-heme uptake is not only a mechanism to bring iron into the cell but also heme and the required amino acids (from the albumin) which the cells need to proliferate

and/or to produce cytokines. The need for HO-1 activity to see HSA-heme induced proliferation further underlines that heme-degradation occurs.

We now also demonstrate that HSA-heme-mediated cell proliferation can be blocked with iron-chelator 311, which is cell membrane permeable, but not with EDTA, which acts extracellularly (Figure 5b).

This information is now also given in the text (page 5, line 28-30 – page 6, line 1-6).

2) It is not clear why free heme does not stimulate cell proliferation (suppl. Fig. 3). Heme is known to easily permeate cells and release iron following its degradation by HO-1, which in turn modulates cellular iron metabolism (for instance, see PMID 1992460). The effects of free heme on cell iron status should be measured as discussed above.

We believe that free heme does not stimulate cell proliferation, because the cells have no protein/amino acid source in our cell culture system. Since albumin is the only protein/amino acid source for the cells in our protein-free system albumin, there is obviously a strong bias and need in the cells to degrade and use albumin as nutrient.

3) To support the claim that TfR1 operates as a receptor for heme-BSA, the authors should perform direct binding assays and determine the affinity constant.

We have performed binding assays of HSA-heme and have now also determined the binding capacity. The dissociation constant (K_d) value of HSA-heme binding is $7,52 \times 10^{-7} M$, which is lower range of what has been reported for transferrin. The K_d for bound diferric transferrin ranges from $10^{-7} M$ to $10^{-9} M$ at physiologic pH, depending on the species and tissue. The K_d of monoferric transferrin is $\sim 10^{-6} M$. This information is now also presented in the Discussion (page 8, line 4-7).

Minor issues:

1) Did BSA used in Fig. 1c contain heme?

We have not determined the heme content of BSA used in this study. Several studies have shown that BSA molecules carry heme, although not so frequent as human albumin (Lee et al. 2007, Monzani et al. 2001, References 37 and 38 in the paper). Nevertheless, it seems as if the albumin/CD71 interaction is species specific. This conclusion is supported not only by HSA

vs BSA discrepancy but also from our studies with murine cells where the HSA-heme is obviously not used as iron source (Figure 2a, 2b). This concept is now also discussed in the new version of our paper (page 8, line 10-16).

2) How much was the expression of transfected human TfR1 (wild type or mutant) compared to endogenous in the murine cells? Did transfected human TfR1 retain the iron responsive elements (IREs) in its mRNA?

The expression levels of human TfR1 in murine Bw-cells was higher compared to the endogenous levels. The human CD71 should have retained the IREs in its mRNA but this was not examined.

Wild type, murine splenocytes, used in this study, do not express human CD71. Murine splenocytes (CD71-mut) were transfected with a mutated, non-functional version of human CD71.

3) Do HO-1 inhibitors (such as tin protoporphyrin IX) block the stimulating effects of heme-BSA in Jurkat cells and the other human cell lines?

We have now used protoporphyrin IX to block HO-1. The results presented in our new Figure 5a demonstrate that the drug inhibits the HSA-heme mediated proliferation but not proliferation in the presence of FCS (control). This information is now also given in the text on page 6, line 1-4.

4) Biliverdin is not “a protoporphyrin without iron”, it is a degradation product of heme, without intact protoporphyrin ring.

This has been corrected in the new version of the manuscript (page 4, line 7). Moreover, we performed an additional control and demonstrate in our new Supplementary Figure 3 that loading of HSA with protoporphyrin IX does not mediate cell proliferation.

5) NO is not a downstream product of the HO-1 reaction; the only downstream gaseous product is CO.

We are sorry for the mistake. Heme degradation is not a source of NO. The sentence on page 6, line 9 has been corrected in the revised version of the paper.

6) *The first sentence of the introduction is not absolutely correct. While iron is essential for the vast majority of known forms of life, there are some minor exceptions such as lactobacilli or Borrelia burgdorferi (see PMID 9269745 and 10834845).*

We are sorry for this incorrectness and we have changed the text accordingly (page 3, line 1).

Reviewers' comments:

Reviewer #1 (Remarks to the Author):

The authors have made extensive changes in response to the reviewers comments, which have improved the manuscript. The additional data help to resolve some, but not all, of the concerns of the reviewers. Nevertheless, the manuscript and the data remain difficult to appreciate – in part because of the organization of the research that includes a plethora of cell types. Furthermore, in several places, what comes across is a lack of rigor when key details are not included in the appropriate place – which may be due in part to the need for some help with editing.

Some examples may be helpful because both the body of the manuscript and data presentation require extensive revision.

Given the considerable revision and additional data the Abstract, which was not changed, needs careful updating. What exactly are the key novel findings? Is one purpose to stress that the transferrin receptor is a scavenger receptor or that it is important in iron metabolism in hemolytic states? Currently, this is not clear. The need for so many different cell types requires further careful justification and organization.

Unfortunately, inconsistent and imprecise statements abound: Not investigating iron transport into human cells, as stated, but heme uptake/transport from heme-albumin. Evidence supports that heme-HSA provide cells with iron from heme catabolism

Another confusing statement - referring to heme as iron through a large amount of iron can be handled by HSA.

P 5 line 26 is iron transferrin meant not transferrin... is inhibited by CD71 mAb?

Intro p3 line 15: Heme simply IS iron-protoporphyrin IX

The data are not clearly presented. For example in Fig 1 and many other figures - by convention the concentration increases from left to right of x-axis not as presented here (high to low –left to right). These must be changed.

Re. the internalization of heme-HSA, Fig 3 needs a higher magnification and additional data to identify the subcellular location of the albumin.

The data in supplementary fig 9 and 10 are a very important part of this research and should be included in the manuscript as regular figures.

P 6 it is stated that EDTA acts extracellularly. What is the evidence that EDTA does not enter cells?

Re. fig 4 are the concentrations of HSA-heme and FAC equivalent in terms of iron because the cells respond differently.

Fig 5b Why are there fewer cells when incubated with HSA-heme than with FCS?

Fig 6. Two different scales for the y axes are needed – expand for the DCs.

Reviewer #2 (Remarks to the Author):

In supplementary figure 4, the authors need to compare cell numbers for cells treated with HSA and cells treated with HSA-heme for each treatment concentration. Based upon the means +/- SEMs presented in the panels, it doesn't look like there will be significant differences in cell numbers between HSA- and HSA-heme-treated cells. This would suggest that the effect of HSA-heme on cell proliferation is restricted to specific cell types.

The data in figure 2b don't support the authors' conclusion that CD71 is the receptor for HSA-heme. If the authors included a cell line that expressed wild-type human CD71 receptor, and saw HSA-heme-induced proliferation, that would support their conclusion.

The authors refer to supplementary figure 9 when describing how anti-CD71 mAbs inhibit cell proliferation, but this figure only shows expression analysis of CD71 receptor.

Is there a legend for figure 3c?

Figure 3f is described in reference to "internalization of HSA-heme into Jurkat cells and a comparison with transferrin", but only HSA internalization is shown in figure 3f. Shouldn't transferrin internalization be shown as well?

The x-axis labels on panels in figure 3g can be revised. It currently reads as if HSA-heme was only included in the first bar, not all the bars. The same comment applies to figure 5.

The data in figure 4 would be more convincing if YK01 cells transfected with HO-1 expression constructs could proliferate in presence of HSA-heme. There still could be a lot of differences between OTHAKA and YK01 cells in addition to HO-1 levels.

Why are there no error bars on the medium controls in figure 4c?

Westerns are needed in figure 4c in addition to the mRNA data already shown. FAC treatment should impact ferritin protein levels.

In figure 5, why are 'control' and 'HSA-heme' groups shown on separate panels? Were they done as separate experiments? If you look at the cell numbers, it looks like they are lower for HSA-heme-treated groups than control groups.

Please replace the heat maps in figure 6a with bar graphs. Also, include statistical analysis of the results. It looks like both HSA and HSA-heme have similar effects on AP1, NFAT, and NfκB activity.

Figure 7c needs a legend.

I believe the post-hoc test is called Tukey's, not Turkey's.

Figure 5 is entitled "HSA-heme is used as iron source". The data in this figure only refer to cell numbers, not iron levels. Without data on iron levels in treated cells, one can't conclude that HSA-heme is used as an iron source.

Reviewer #3 (Remarks to the Author):

The revised manuscript is improved and has addressed most points raised by reviewers. There are, however, a couple of remaining issues.

1) The authors tried to show that heme-HSA promotes cellular iron loading. However, the data in Fig. 4c are weak, mainly because of the poor experimental design. The TFR1 mRNA response is expected

and this is fine. The lack of ferritin mRNA regulation is likewise expected, because ferritin expression is regulated translationally; therefore, only measuring ferritin protein levels would be meaningful. The regulation of IRP1 mRNA is unexpected and not supported by previous literature; it is well established that IRP1 is regulated post-translationally. The authors should provide one additional piece of evidence that cells are iron-loaded apart from TFR1 mRNA. As indicated in my original review, this can be done for instance by measuring ferritin expression (by Western), IRP1 and IRP2 activities (by EMSA), IRP2 expression (by Western), or levels of labile iron (by a fluorescent dye). It would also be nice to use free heme as additional positive control for iron loading.

2) The authors argue that “free heme does not stimulate cell proliferation, because the cells have no protein/amino acid source” in their cell culture system. How can they explain the stimulatory effects of FAC in Fig. 2c?

Response to the Reviewers

Reviewer #1:

We thank the reviewer for the careful evaluation of our work. We have now edited and revised the manuscript according to the proposals and comments of the reviewers and to present key details more rigor. Our paper provides a plethora of new data and most of the experiments were performed with one cell line, Jurkat T cells. We appreciate to confirm our key findings in a diversity of cell lines and model systems because it strengthens our observations and conclusions.

To the specific points raised by you we would like to respond as follows:

Reviewer

Given the considerable revision and additional data the Abstract, which was not changed, needs careful updating. We have followed your suggestion and updated the Abstract with the additional data provided in our manuscript. We interpret our findings that CD71 is both, a scavenger receptor and important in iron metabolism.

The need for so many different cell types requires further careful justification and organization.

In order to improve the organization of our paper, we have now edited the description of the figures. So, we hope that the reader can easily identify the cell type used to obtain the data.

Unfortunately, inconsistent and imprecise statements abound: Not investigating iron transport into human cells, as stated, but heme uptake/transport from heme-albumin. Evidence supports that heme-HSA provide cells with iron from heme catabolism

The inconsistent and imprecise statements concerning the usage of iron transport vs HAS-heme uptake has been now corrected in the Abstract and throughout the text.

Another confusing statement - referring to heme as iron through a large amount of iron can be handled by HSA. This statement has been changed (page 3, line 25).

P 5 line 26 is iron transferrin meant not transferrin... is inhibited by CD71 mAb?

We have changed this impreciseness (page 5, line 23).

Intro p3 line 15: Heme simply IS iron-protoporphyrin IX

The description of heme has been changed in the new version (page 3, line 15).

The data are not clearly presented. For example in Fig 1 and many other figures - by convention the concentration increases from left to right of x-axis not as presented here (high to low –left to right). These must be changed.

We have now changed the presentation of Figure 1a-e, 2b, 3g, 4a, 5a-c, 6a, 7a with increasing concentrations from left to right.

Re. the internalization of heme-HSA, Fig 3 needs a higher magnification and additional data to identify the subcellular location of the albumin.

We have reformatted the figure in the new version of our paper. The fate of albumin in the cell is of course of interest but needs to be studied in future studies and under conditions (e.g. in the presence of serum) where cells will not need to degrade HSA for nutrition.

The data in supplementary fig 9 and 10 are a very important part of this research and should be included in the manuscript as regular figures.

We agree with you that both figures show important parts of our research. Yet, in Supplementary Figure 9 we show the control staining for the CD71 mAbs used in this study and Supplementary Figure 10 demonstrates the inhibitory effect of CD71 mAbs on HSA-heme triggered proliferation in a different method as shown in Figure 2 in the regular figures. Since the amount of information, data and figures in our paper

is already pretty high and since we have now added 2 additional figures (Figure 5a and 5d) we prefer to present both figures in the supplementary part.

P 6 it is stated that EDTA acts extracellularly. What is the evidence that EDTA does not enter cells?

EDTA is a well-known chelator and acts, according to published data, mainly extracellularly. This information is now also given in the text on page 6, line 5.

Re. fig 4 are the concentrations of HSA-heme and FAC equivalent in terms of iron because the cells respond differently.

The concentrations in terms of iron are indeed differently between HSA-heme and FAC. The concentrations (HSA-heme 200 µg/ml; FAC 25 µg/ml) were the same as in the proliferation assays presented in Figure 2c. At the used concentrations, the amount of iron was 251 µg/ml and 5 µg/ml for HSA-heme and FAC, respectively. This information is now also given in the paper on page 12, line 5.

Fig 5b Why are there fewer cells when incubated with HSA-heme than with FCS?

As described in our paper, culture of cells with HSA-heme means that there were no of other proteins present in the medium. In contrast to FCS, we always find lower proliferation with HSA-heme alone in the medium compared to RPMI medium supplemented with 10% FCS (e.g. in Figure 2a and 2c).

Fig 6. Two different scales for the y axes are needed – expand for the DCs.
The cytokine data of the immature DCs is now presented in a separated figure with a different scaling of the y-axis.

Reviewer #2:

We thank the reviewer for the careful evaluation of our work and the constructive critique.

To the points raised by you we would like to respond as follows:

Reviewer:

In supplementary figure 4, the authors need to compare cell numbers for cells treated with HSA and cells treated with HSA-heme for each treatment concentration. Based upon the means +/- SEMs presented in the panels, it doesn't look like there will be significant differences in cell numbers between HSA- and HSA-heme-treated cells. This would suggest that the effect of HSA-heme on cell proliferation is restricted to specific cell types.

We have tested the impact of HSA-heme on different cell lines. In comparison with plasma-derived HSA we have observed that cells need less HSA-heme to proliferate. This is also the case for TF1 cells. TF1 cells are erythroid cells which start to differentiate in response to heme towards erythrocyte-lineage. Such a differentiation process is likely to be accompanied with a reduced proliferation rate in TF1 cells. The role of HSA-heme on the proliferative response of other cell types needs to be tested in more detail. This is now also mentioned in the text on page 7, line 30.

The data in figure 2b don't support the authors' conclusion that CD71 is the receptor for HSA-heme. If the authors included a cell line that expressed wild-type human CD71 receptor, and saw HSA-heme-induced proliferation, that would support their conclusion.

This is correct. Data in Figure 2a, 2c as well as Figure 3a, b, c support that CD71 is a receptor for HSA-heme. Fig. 2b is only shown to demonstrate that a defect CD71 is not sufficient to promote proliferation by HSA-heme. This information is now also made clearer in the legend to the figure.

The authors refer to supplementary figure 9 when describing how anti-CD71 mAbs inhibit cell proliferation, but this figure only shows expression analysis of CD71 receptor.

We are sorry for this mistake. It is actually described in Supplementary Figure 10. This has been now changed in the text (page 5, line 1).

Is there a legend for figure 3c?

We have now added a legend for Figure 3c in addition to the description in the legend to Figure 3.

Figure 3f is described in reference to "internalization of HSA-heme into Jurkat cells and a comparison with transferrin", but only HSA internalization is shown in figure 3f. Shouldn't transferrin internalization be shown as well?

We compare the quantitative uptake of HSA-heme and iron-loaded transferrin in Figure 3e. The pictures for transferrin uptake are not available but have been shown in several publications from other groups before. It will be interesting however, to compare the route of HSA-heme and transferrin intracellularly and particularly in the culture conditions where proteins may not be required for the cells to be degraded.

The x-axis labels on panels in figure 3g can be revised. It currently reads as if HSA-heme was only included in the first bar, not all the bars. The same comment applies to figure 5.

The x-axes of both figures have been corrected in the new version of our paper.

The data in figure 4 would be more convincing if YK01 cells transfected with HO-1 expression constructs could proliferate in presence of HSA-heme. There still could be a lot of differences between OTHAKA and YK01 cells in addition to HO-1 levels.

YK01 is an EBV-transformed lymphoblastoid B cell line from a patient suffering with HO-1 deficiency. OTHAKA is an EBV-transformed lymphoblastoid B cell line generated from a healthy donor. You are right that these 2 different cell lines may nevertheless have additional molecular differences. This has not been analyzed so far. Yet, CD71 and iron-uptake are obviously intact since YK01 cells proliferate in response to transferrin containing medium (FCS). It is intriguing that there are also other molecular differences between both cell lines. However, we provide more direct evidence that HO-1 is important for HSA-heme catabolism and HSA-heme induced proliferation. Results presented in Figure 4c demonstrate that proliferation of Jurkat cells in the presence of HSA-heme but not FCS is inhibited by Tin Protoporphyrin, an inhibitor of HO-1.

Why are there no error bars on the medium controls in figure 4c?

We now present the data in Figure 5b (former figure 4c) as fold change in order to make the comparison with the medium control values easier.

Westerns are needed in figure 4c in addition to the mRNA data already shown. FAC treatment should impact ferritin protein levels.

We have now analyzed the regulation of CD71 and ferritin at the protein level by intracellular staining and analyses via flow cytometry. We were not able to test the expression of IRP1 and IRP2, since we could not buy and obtain the antibodies before the end of June. The results are presented in our new Figure 5d demonstrate that HSA-heme down-regulates CD71 and upregulates ferritin expression. This information is now also given in the text on page 6, line 10.

We have now also analyzed the levels of intracellular labile iron in Jurkat T cells in the presence of HSA-heme versus FAC and found that HSA-heme increases the amounts of intracellular iron (Figure 5a). This information is now also presented in the text (page 6, line 1).

In figure 5, why are 'control' and 'HSA-heme' groups shown on separate panels? Were they done as separate experiments? If you look at the cell numbers, it looks like they are lower for HSA-heme-treated groups than control groups.

The control and HSA-heme groups in Figure 5 are now in the same panels.

Please replace the heat maps in figure 6a with bar graphs. Also, include statistical analysis of the results. It looks like both HSA and HSA-heme have similar effects on AP1, NFAT, and NfκB activity.

We have now replaced the heat maps with bar graphs and show statistical analysis of the data presented in Figure 6.

Figure 7c needs a legend.

We have now added a legend for Figure 8c (Figure 7c in the old version) in addition to the description in the legend to Figure 8.

I believe the post-hoc test is called Tukey's, not Turkey's.

You are right, we are sorry for the mistake and have corrected it throughout the paper.

Figure 5 is entitled "HSA-heme is used as iron source". The data in this figure only refer to cell numbers, not iron levels. Without data on iron levels in treated cells, one can't conclude that HSA-heme is used as an iron source. We now show in our new Figure 5a, that HSA-heme treatment of Jurkat cells increases the intracellular of the labile iron. We conclude that HSA-heme is used as an iron source because it is the only potential molecular complex with iron in the culture system and we can inhibit the effect of HSA-heme with iron-chelators. The title for Figure 5 has been changed accordingly.

Reviewer #3:

We thank the reviewer for the careful evaluation of our work and the constructive critique.

To the points raised by you we would like to respond as follows:

Reviewer:

The revised manuscript is improved and has addressed most points raised by reviewers. There are, however, a couple of remaining issues.

*1) The authors tried to show that heme-HSA promotes cellular iron loading. However, the data in Fig. 4c are weak, mainly because of the poor experimental design. The TFR1 mRNA response is expected and this is fine. The lack of ferritin mRNA regulation is likewise expected, because ferritin expression is regulated translationally; therefore, only measuring ferritin protein levels would be meaningful. The regulation of IRP1 mRNA is unexpected and not supported by previous literature; it is well established that IRP1 is regulated post-translationally. The authors should provide one additional piece of evidence that cells are iron-loaded apart from TFR1 mRNA. As indicated in my original review, this can be done for instance by measuring ferritin expression (by Western), IRP1 and IRP2 activities (by EMSA), IRP2 expression (by Western), or levels of labile iron (by a fluorescent dye). It would also be nice to use **free heme** as additional positive control for iron loading.*

We have now analyzed the regulation of CD71 and ferritin at the protein level by intracellular staining and analyses via flow cytometry. We were not able to test the expression of IRP1 and IRP2, since we could not buy and obtain the antibodies before the end of June. The results are presented in our new Figure 4d demonstrate that HSA-heme down-regulates CD71 and upregulates ferritin expression. This information is now also given in the text on page 6, line 10.

We have now also analyzed the levels of intracellular labile iron in Jurkat T cells in the presence of HAS-heme versus FAC and found that HSA-heme increases the amounts of intracellular iron (Figure 5a). This information is now also presented in the text (page 6, line 1).

2) The authors argue that “free heme does not stimulate cell proliferation, because the cells have no protein/amino acid source” in their cell culture system. How can they explain the stimulatory effects of FAC in Fig. 2c?

Because FAC was added to cells cultured in FCS-supplemented medium. In order to make this point clearer we have now changed the legend to the figure and it's description.

REVIEWERS' COMMENTS:

Reviewer #2 (Remarks to the Author):

Authors have addressed issues raised in previous review.

Reviewer #3 (Remarks to the Author):

The revised manuscript is improved and all major issues have been addressed